# Aberrant interaction of FUS with the U1 snRNA provides a molecular mechanism of FUS induced amyotrophic lateral sclerosis

Daniel Jutzi [1,8], Sébastien Campagne [2,8], Ralf Schmidt[3], Stefan Reber[1], Jonas Mechtersheimer[1], Foivos Gypas [3,4], Christoph Schweingruber [5], Martino Colombo[6], Christine von Schroetter[2], Fionna E. Loughlin[2,7], Anny Devoy [1], Eva Hedlund [5], Mihaela Zavolan [3], Frédéric H.-T. Allain[2,9✉] & Marc-David Ruepp [1,9✉]

Mutations in the RNA-binding protein Fused in Sarcoma (FUS) cause early-onset amyotrophic lateral sclerosis (ALS). However, a detailed understanding of central RNA targets of FUS and their implications for disease remain elusive. Here, we use a unique blend of crosslinking and immunoprecipitation (CLIP) and NMR spectroscopy to identify and characterise physiological and pathological RNA targets of FUS. We find that U1 snRNA is the primary RNA target of FUS via its interaction with stem-loop 3 and provide atomic details of this RNA-mediated mode of interaction with the U1 snRNP. Furthermore, we show that ALS-associated FUS aberrantly contacts U1 snRNA at the Sm site with its zinc finger and traps snRNP biogenesis intermediates in human and murine motor neurons. Altogether, we present molecular insights into a FUS toxic gain-of-function involving direct and aberrant RNA-binding and strengthen the link between two motor neuron diseases, ALS and spinal muscular atrophy (SMA).

[1] United Kingdom Dementia Research Institute Centre, Institute of Psychiatry, Psychology and Neuroscience, King's College London, Maurice Wohl Clinical Neuroscience Institute, London, UK. [2] Institute of Molecular Biology and Biophysics, Department of Biology, ETH Zürich, CH-8093 Zürich, Switzerland. [3] Computational and Systems Biology, Biozentrum, University of Basel, CH-4056 Basel, Switzerland. [4] Friedrich Miescher Institute for Biomedical Research, CH-4058 Basel, Switzerland. [5] Department of Neuroscience, Karolinska Institutet, 171 77 Stockholm, Sweden. [6] Celgene Institute of Translational Research (CITRE), 41092 Seville, Spain. [7] Present address: Monash Biomedicine Discovery Institute and Department of Biochemistry and Molecular Biology, Monash University, Melbourne, VIC 3800, Australia. [8] These authors contributed equally: Daniel Jutzi, Sébastien Campagne. [9] These authors jointly supervised this work: Frédéric H.-T. Allain, Marc-David Ruepp. ✉email: allain@bc.biol.ethz.ch; marc-david.ruepp@kcl.ac.uk

RNA processing is an essential part of gene expression, and disturbed RNA metabolism has been linked to several neurodegenerative diseases[1]. Among these diseases is amyotrophic lateral sclerosis (ALS), a relentless adult-onset disease characterised by loss of motor neurons in the motor cortex and spinal cord, leading to muscle weakness and eventually paralysis and death[2]. Mutations in the *FUS* gene are at the origin of FUS-linked ALS, and some are associated with a particularly aggressive disease phenotype with juvenile onset[3,4]. Insoluble FUS inclusions in neurons and glial cells represent the pathological hallmark of this type of disease. FUS pathology is also observed in frontotemporal dementia (FTD), where FUS co-aggregates with the other members of the FET-family, EWS and TAF15, as well as with their nuclear import receptor β2/transportin-1 (TNPO1)[5,6]. In this case, aggregation occurs in the absence of *FUS* gene mutations and may be caused by aberrant post-translational modifications[7].

The FUS protein comprises two functional modules. A conserved N-terminal region of low complexity (LC) consisting of QGSY-rich and G-rich domains drives liquid–liquid phase separation (LLPS) and mediates protein–protein interactions[8]. In contrast, the C-terminal module is responsible for nucleic acid binding and contains two globular RNA-binding domains (RBD), an RNA-recognition motif (RRM) and a zinc finger (ZnF), each embedded in RGG-rich sequences[9]. The last 29 amino acids constitute a PY-type nuclear localisation signal (NLS)[10]. As a ubiquitously expressed, predominantly nuclear protein, FUS may regulate gene expression at different levels. Considerable evidence links FUS to the splicing machinery, especially to the U1 snRNP and U11/U12 di-snRNP[11,12]. However, few aspects of these interactions are known. Consistent with a role as splicing factor, loss of FUS induces widespread splicing alterations, affecting both U2-type and U12-type introns[12,13]. While we previously reported that the LC domain of FUS is sufficient to recruit the U12-type spliceosome, how FUS regulates U2-type splicing remains elusive.

ALS-associated mutations typically disrupt the nuclear localisation signal of FUS, leading to cytoplasmic mislocalisation and eventually the formation of aggregates[10]. Recent mouse models strongly suggest that FUS causes motor neuron degeneration through a cytoplasmic toxic gain-of-function, although a reduction of nuclear FUS could contribute to the disease[14–16]. RNA binding is required for full FUS toxicity in various ALS models[17,18]. Hence, a detailed characterisation of both cytoplasmic and nuclear RNA interaction networks of FUS is not only key to better understanding its physiological function but could also provide valuable insights into the molecular mechanisms underlying neurodegeneration in ALS.

In this study, we perform a combination of CLIP experiments to identify physiological as well as pathological RNA targets of FUS. We identify the spliceosomal U1 snRNA as the major FUS target in the nucleus and provide the solution structure for this RNA-mediated mode of U1 snRNP recognition: The RRM of FUS contacts stem-loop 3 (SL3) of the U1 snRNA, which protrudes from the globular core of the particle. Furthermore, we show that in ALS models, FUS aberrantly interacts with the U1 snRNA in the cytoplasm, leading to impaired snRNP biogenesis. These findings provide insights into the mechanism of FUS-dependent splicing regulation and suggest that impaired snRNP biogenesis molecularly links the motor neuron diseases ALS and SMA.

## Results

**An RBD-centric FUS CLIP approach.** We performed CLIP-Seq[19] with three different FUS constructs to comprehensively identify direct RNA targets of FUS on a transcriptome-wide scale (Fig. 1a). Besides the full-length protein, we employed a FUS construct comprising only its RNA-binding module (amino acids 242–526) to study the importance of the LC region (aa 1–241), which enables FUS to form complexes with other hnRNPs (Fig. 1b and Supplementary Fig. 1a, b). Such cofactors can modulate the binding landscape of an RNA-binding protein in vivo[20]. Finally, we aimed at identifying RNA targets that could be implicated in neurodegeneration by performing CLIP with cytoplasmic ΔLC-FUS harbouring the ALS-associated P525L mutation combined with a heterologous nuclear export signal (NES). All FUS constructs were Twin-Strep tagged and stably introduced in SH-SY5Y neuroblastoma cells by lentiviral transduction. This approach allowed us to purify FUS–RNA complexes to near homogeneity, which reduces the risk of false positives introduced by contaminating RNA-binding proteins[21]. The transgenes were expressed close to physiological levels (Fig. 1c) and in the background of a *FUS* knockout (KO) to prevent competition for binding sites with endogenous FUS. Immunofluorescence confirmed the expected nuclear and cytoplasmic localisation, respectively (Fig. 1d–f and Supplementary Fig. 1c). The CLIP experiments were performed in triplicate each and a no-cross-link control was included to monitor the specificity of the signal. As expected, the autoradiographs revealed strictly cross-link-dependent protein–RNA complexes migrating slightly slower than the free FUS constructs. RNA was isolated from the regions indicated by the red dashed lines and converted into cDNA libraries for high-throughput sequencing. To normalise the CLIP data to the input, we profiled the transcriptomes of all CLIP cell lines by RNA sequencing after depletion of ribosomal RNA.

**CLIP-Seq reveals the U1 snRNA as a major FUS target.** In agreement with previous CLIP studies, we found that the binding signatures of full-length FUS were evenly distributed along the entire length of transcripts with around 70% of the reads mapping to introns (Fig. 2a and Supplementary Fig. 2a), reflecting binding of FUS to pre-mRNAs[13,22]. In contrast, the percentage of intronic reads was reduced to ~40% in the ΔLC-FUS CLIP, suggesting that the LC domain promotes co-transcriptional binding of FUS to introns. This agrees with our recent finding that liquid–liquid phase separation by FUS is required for its association with chromatin[23]. In particular, the first intron of the hnRNPA2B1 pre-mRNA displays binding that is heavily dependent on the LC domain of FUS (Fig. 2b). Nevertheless, loss of this unstructured domain does not alter the widespread nature of FUS binding to exonic regions as shown for hnRNPA2B1 pre-mRNA or in the long non-coding RNA MALAT1, indicated by the similar CLIP peak distributions (Fig. 2b).

To define FUS-binding sites at single-nucleotide resolution, we exploited a characteristic feature of CLIP-Seq data: peptide remnants that remain cross-linked to the RNA cause the reverse transcriptase to introduce deletions in the library preparation step, thereby allowing for the identification of cross-link sites at single-nucleotide resolution[24]. Indeed, deletions represent the most common type of mutation in our data (Supplementary Fig. 2b). Hence, we looked for genomic positions where deletions are supported by at least three independent reads and overall mutation frequency was below 0.5 to exclude false positives due to single-nucleotide polymorphisms (see scheme in Fig. 2c). After clustering of proximal deletions (within a range of ten nucleotides) and only considering deletions identified in all biological replicates, we defined a set of 14,327 highly reproducible FUS-binding sites. These binding sites were preferentially located in single-stranded regions flanked by sequences with increased folding propensity

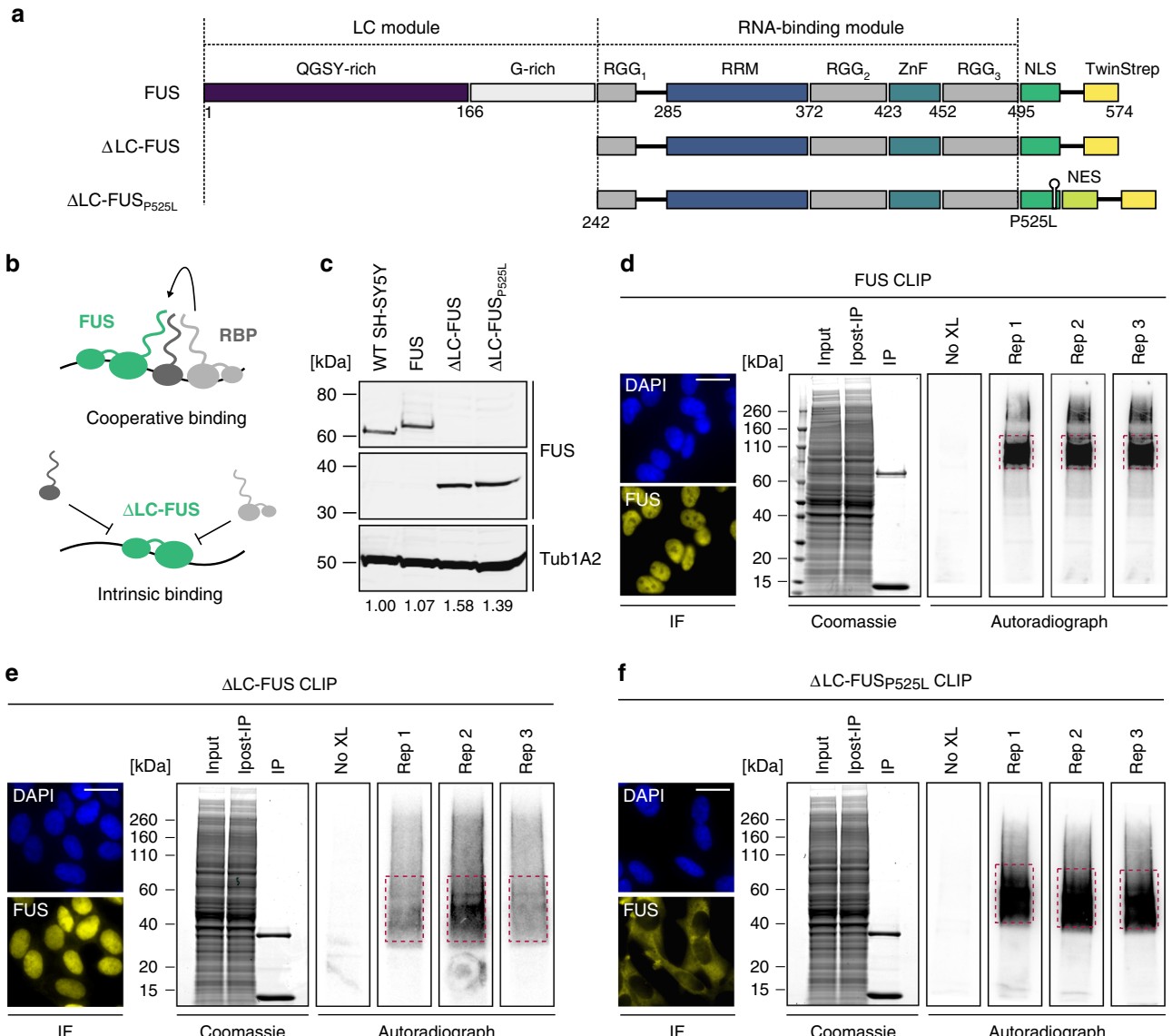

**Fig. 1 An RBD-centric FUS CLIP approach. a** Schematic representation of the Twin-Strep-tagged FUS constructs used for CLIP. **b** In addition to binding as a free molecule, FUS may be recruited to RNA via interactions with other RBPs. This may confer foreign specificity to FUS and promote binding to low-affinity binding sites. **c** Western blot showing expression of FUS CLIP constructs in comparison to endogenous FUS in the parental SH-SY5Y cell line, detected using anti-FUS antibodies. Tub1A2 served as a loading control. Relative FUS expression levels were determined by densitometry. $n = 3$. **d–f** Immunofluorescence confirmed the expected nuclear or cytoplasmic localisation of FUS CLIP constructs, visualised using anti-FUS antibodies. Nuclei were counterstained with DAPI. Scale bar $= 10\,\mu m$, $n = 1$. Protein–RNA complexes were purified using Strep-Tactin beads, separated by SDS-PAGE and their purity was assessed by Coomassie staining. Autoradiographs of three biological replicates reveal characteristic shifts above the free protein. RNA was isolated from the regions of the membranes indicated by red dashed lines.

(Fig. 2d), consistent with the recently described specificity of the FUS RRM for stem-loop structures[9]. To normalise the CLIP data to input RNA-Seq data, we defined 200-nucleotide windows around the cross-link sites and calculated the ratio of CLIP-Seq reads to RNA-Seq reads (Fig. 2c). This analysis yielded significantly overlapping sets of highly enriched transcripts for the three biological replicates (Supplementary Fig. 2c). Again, consistent with the FUS-binding specificity, we found that spliceosomal snRNAs, which are rich in stem-loops, were significantly more enriched relative to messenger and long non-coding RNAs (Fig. 2e). Among the snRNAs, the U1 snRNA was the top target and represented by far the most enriched transcript in the whole dataset (Fig. 2f), confirming previous results that linked FUS to the U1 snRNP[11,25,26] and suggesting that the contact occurs via the snRNA. Using RNA–RNP

immunoprecipitation (RIP), we found that FUS, but not an RNA-binding deficient FUS mutant, selectively interacts with the U1 snRNA, confirming that FUS exclusively contacts the RNA component of the U1 snRNP (Fig. 2g). In contrast, the interaction with the U11/U12 di-snRNP is RNA-independent, as the U11 snRNA co-purified with both constructs. This is in agreement with our previous study showing that the LC domain of FUS is sufficient to promote U12-type intron splicing and suggests that FUS may have evolved distinct mechanisms to regulate the splicing of U2-type and U12-type introns[12]. We then mapped the precise FUS-binding site on U1 snRNA by computing the number of cross-link-induced deletions for every nucleotide position along the U1 snRNA primary sequence (Fig. 2h). The resulting binding footprint revealed a distinct cross-link cluster in stem-loop 3 (SL3), arguing that

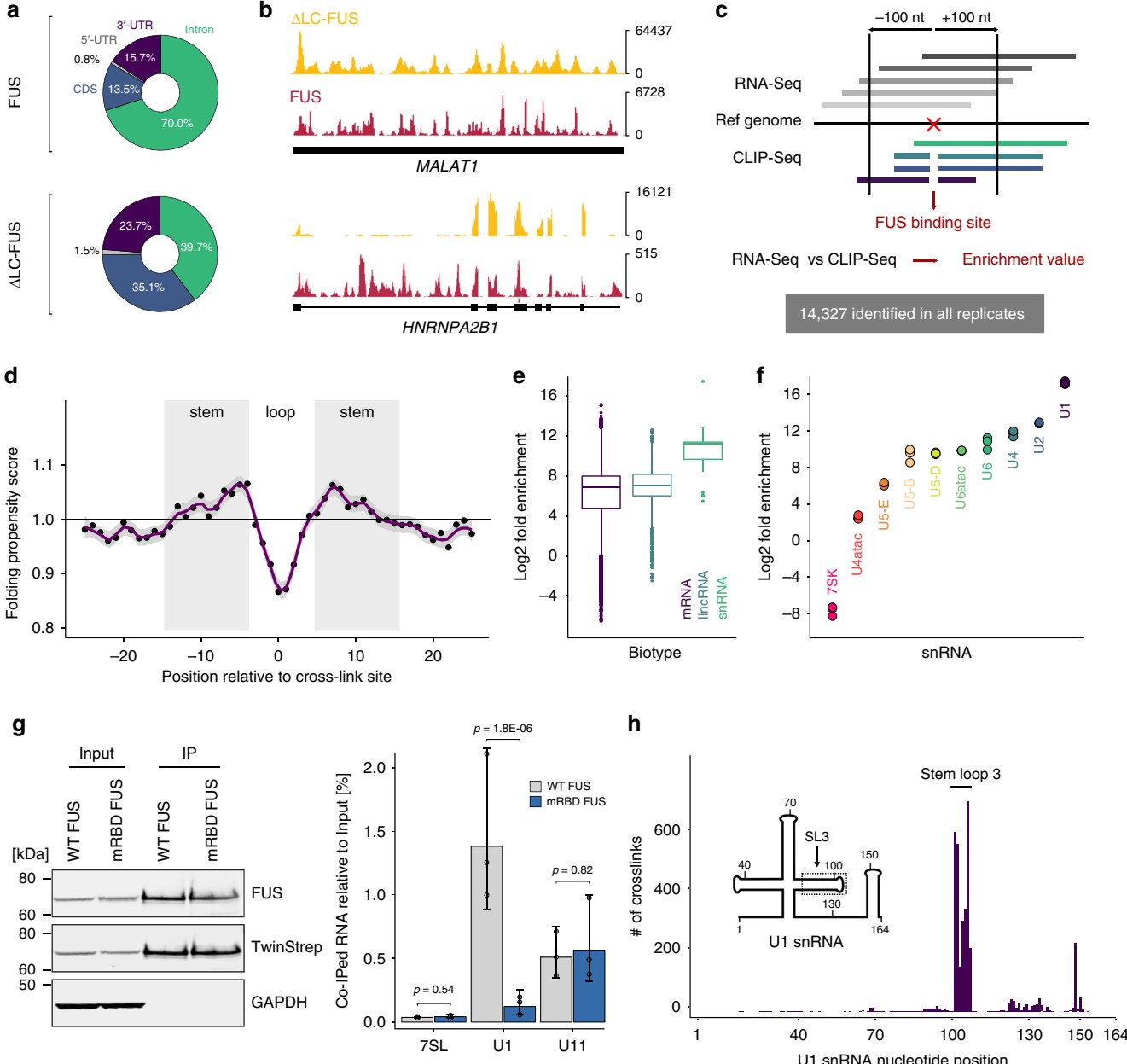

**Fig. 2 CLIP-Seq reveals U1 snRNA as a major FUS target. a** Read distribution of FUS and ΔLC-FUS CLIP experiments in pre-mRNAs according to the annotated transcript region. **b** Genome browser views showing the distribution of FUS and ΔLC-FUS CLIP reads on selected genes. **c** Schematic representation of the analysis approach used to infer FUS-binding sites from CLIP data. **d** Folding propensity of nucleotides proximal to cross-link sites identified in all biological replicates. The error band depicts the standard error of the local regression curve. $n = 3$. **e** Box plot showing the distribution of enrichment values for binding sites located in distinct transcript biotypes. The plot displays median lines, interquartile range (IQR) boxes, 1.5 × IQR whiskers and remaining data points. $n = 4929$ (lincRNA), 115,104 (mRNA), 125 (snRNA), examined in three biological replicates. **f** Dot plot showing the mean enrichment value inferred from all binding sites of individual snRNAs (corresponding to panel e) for three biological replicates. **g** WT and RNA-binding-deficient FUS was purified from HeLa cells and detected by western blotting using anti-FUS and anti-Twin-Strep antibodies. GAPDH served as a loading control. Extracts corresponding to $2 \times 10^5$ cell equivalents were loaded as input, IP fractions correspond to $2 \times 10^6$ cell equivalents. RNA levels were assessed by RT-qPCR and quantified relative to the input. Mean values and standard deviations of three biological replicates are shown. $P$ values were computed from log-transformed values using two-sided unequal variance Welch's $t$ test. $n = 3$. **h** Bar plot displaying the number of cross-links for every nucleotide along the U1 snRNA primary sequence.

this structural element represents the major contact point between FUS and U1 snRNP. This interaction site is not only accessible in the previous crystal structure of the U1 snRNP[27] but also in the recent cryo-EM structure of the human spliceosomal pre-B complex[28]. Such a mode of spliceosome recognition was never described and prompted us to further explore the molecular details of this interaction.

**The RRM of FUS contacts the U1 snRNP via stem-loop 3.** To validate the FUS–U1 snRNP interaction in vitro, we reconstituted U1 snRNPs[29] and produced a FUS–RBD construct encompassing the RRM, RGG2 and zinc-finger domains (aa 269–454) (Fig. 3a, b). We observed that FUS–RBD co-migrated with U1 snRNP in analytic size-exclusion experiments (Supplementary Fig. 3a), indicating that the RNA-binding domains of FUS are sufficient to

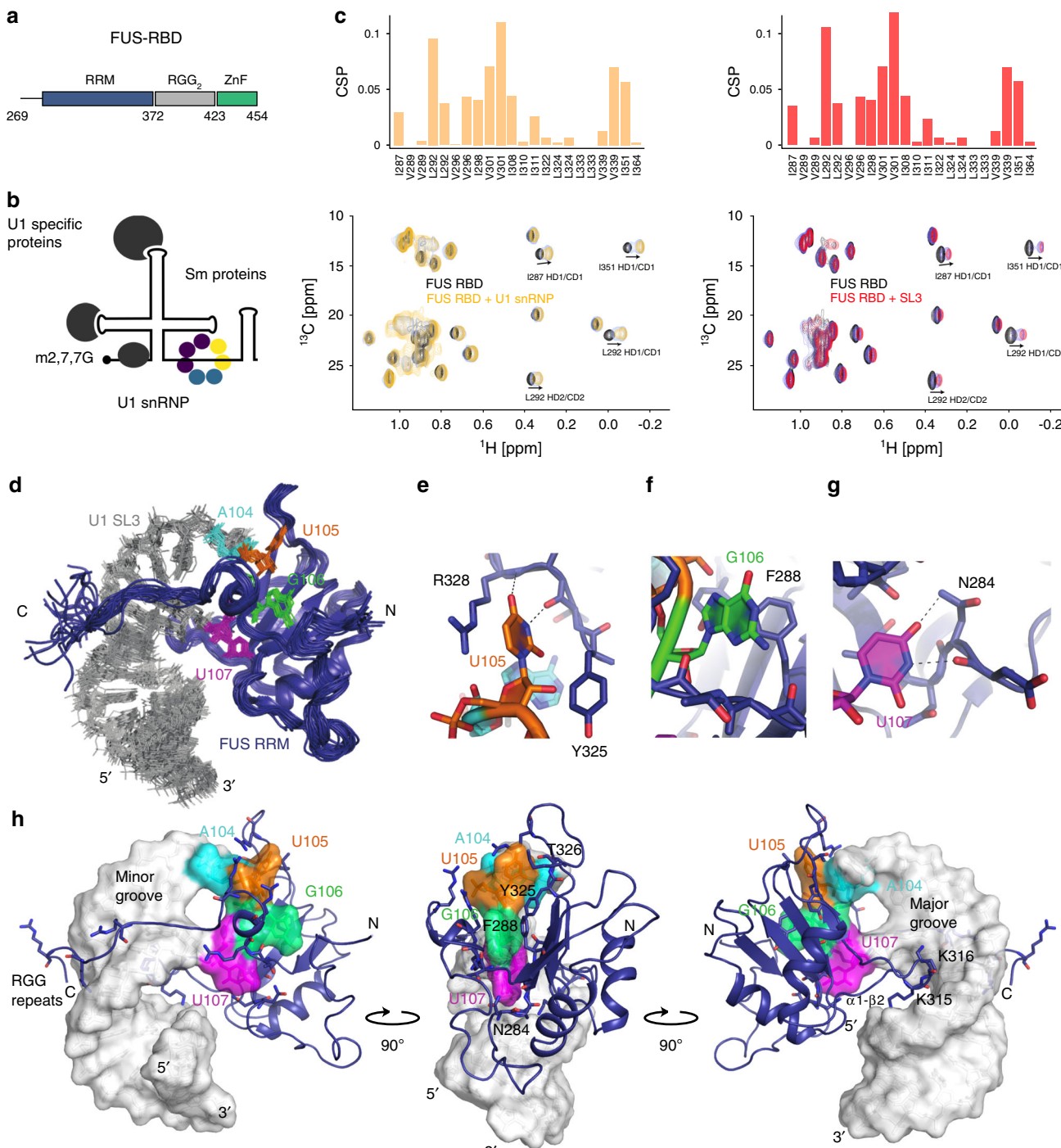

**Fig. 3 Structural basis of the FUS–U1 snRNP interaction. a** Schematic representation of the FUS RBD. **b** Scheme of U1 snRNP. **c** Plot showing the chemical shift perturbations experienced by ILV methyl groups of FUS RBD upon addition of U1 snRNP (orange) or U1 SL3 (red). Below each graph, an overlay of the 2D $^{13}C$-$^1H$ HMQC spectra of the free and bound FUS–RBD is shown. $n = 3$. **d** Ribbon representation of the solution structure of FUS RRM aa 260–390 (blue) bound to U1 SL3 (grey). **e** Close-up view on U105-specific recognition. **f** Close-up view on G106-specific recognition. **g** Close-up view on U107-specific recognition. **h** Lowest energy model of the NMR ensemble. The surface of the RNA is depicted in grey, while the FUS RRM is shown as a ribbon and coloured in blue.

contact the particle. To further delineate the interaction surfaces, we used solution-state NMR spectroscopy and prepared a $^{15}N$, $^{2}H$ and $^{13}C$ ILV-labelled FUS–RBD sample. With this labelling scheme, the NMR signals of ILV methyl groups were monitored upon addition of U1 snRNP on the 2D $^{13}C$-$^1H$ heteronuclear multiple quantum correlation (HMQC) spectrum. Since the zinc finger does not contain any ILV residues, it was followed on a 2D

$^{15}N$-$^1H$ heteronuclear single quantum correlation (HSQC) spectrum. Upon addition of U1 snRNP, we observed that the ILV methyl group signals experienced chemical shift changes, indicating a direct interaction between U1 snRNP and FUS–RBD (Fig. 3c). Strikingly, these shifts were fully reproduced using an isolated SL3 RNA fragment instead of U1 snRNP, confirming that FUS RRM interacts with U1 snRNP on SL3. In contrast, the

**Table 1 NMR ensemble statistics of FUS: U1 SL3 complex.**

| | FUS RRM | U1 SL3 |
|---|---|---|
| *NMR distance and dihedral constraints* | | |
| Distance restraints | | |
| Total NOE | 3569 | 344 |
| Intra-residue | 631 | 132 |
| Inter-residue | 2910 | 192 |
| Sequential ($|i - j| = 1$) | 915 | 138 |
| Nonsequential ($|i - j| > 1$) | 1995 | 54 |
| Hydrogen bonds[a] | 28 | 20 |
| Protein–nucleic acid intermolecular | 92 | |
| Total dihedral angle restraints[b] | | |
| Protein | 66 | |
| φ | 33 | |
| ψ | 33 | |
| Nucleic acid | | 125 |
| Base pair | | |
| Sugar pucker | | 25 |
| Backbone | | 100 |
| Based on A-form geometry | | 125 |
| *Structure statistics* | | |
| Violations (mean and s.d.) | | |
| Distance constraints > 0.3 Å | 3.0 ± 1.2 | |
| Dihedral angle constraints > 5° | 1.1 ± 0.9 | |
| Max. dihedral angle violation (°) | 6.95 ± 3.96 | |
| Max. distance constraint violation (Å) | 0.38 ± 0.03 | |
| Deviations from idealised geometry | | |
| Bond lengths (Å) | 0.0041 ± 0.0001 | |
| Bond angles (°) | 1.404 ± 0.008 | |
| Average pairwise r.m.s. deviation[c] (Å) | | |
| Protein | | |
| Heavy | 0.45 ± 0.06 | |
| Backbone | 0.20 ± 0.04 | |
| RNA | | |
| All RNA heavy | | 0.72 ± 0.16 |
| RNA backbone | | 0.71 ± 0.15 |
| Complex | | |
| Protein and nucleic acid backbone | 0.71 ± 0.16 | |
| Protein and nucleic acid heavy | 0.79 ± 0.15 | |

[a]Hydrogen bond constraints were identified from slow-exchanging amide protons in D$_2$O and imino protons in H$_2$O.
[b]Dihedral angle based TALOS+, sugar puckers based on homonuclear TOCSY, RNA backbone constraints in A form stem based on standard A-form geometry.
[c]Root-mean-squared deviation was calculated using the protein residue range 284-370 (chain ID: A) and the RNA residue range 4-24 (chain ID: B) calculated on the ensemble of 18 NMR structures.

amide NMR signals corresponding to the zinc finger did not show large changes upon addition of U1 snRNP or SL3 but rather broadened, indicating that the zinc finger does not find its preferential binding site on U1 snRNP (Supplementary Fig. 3b).

To further characterise the interaction between FUS and U1 snRNP, we truncated the protein to the region only containing the FUS RRM and RGG2 (aa 260–390, hereafter referred to as FUS RRM). FUS RRM binds to SL3 with a moderate binding affinity of 8 µM (Supplementary Fig. 3c). Upon the formation of the protein–RNA complex in the NMR tube, we observed chemical shift changes mainly occurring at the β-sheet surface of the RRM, at the α1-β2 loop and at the C-terminal extension containing RGG2 (Supplementary Fig. 3d, e). The same experiment was performed using a $^{15}$N–$^{13}$C uniformly labelled sample of SL3 to monitor the changes on the RNA resonances. The NMR signal (H8/H6 and H1′) corresponding to the 3′-part

of the loop (A104-U107) were the most affected (Supplementary Fig. 4a–c).

We next solved the solution structure of FUS RRM bound to SL3 using 3569 and 344 intramolecular nuclear Overhauser effect (NOE)-derived distance constraints for the protein and the RNA, respectively, as well as 94 intermolecular NOEs (Supplementary Fig. 4d). The resulting ensemble of 18 NMR structures overlays with a backbone root-mean-square deviation of 0.79 Å (Fig. 3d and Table 1). Three bases are recognised by the RRM β-sheet surface: (i) U105 forms direct hydrogen bonds with the main chain of T326; (ii) G106 stacks against the side chain of F288 and (iii) U107 establishes direct hydrogen bonds with the main and side chain of N284 (Fig. 3e–g). The analysis of the structure revealed that the RRM recognises the YNY motif located at the 3′-part of the SL3, similarly to our recent structure of FUS RRM bound to a pre-mRNA stem loop[9]. The structure of the protein–RNA complex also shows that the RNA loop is sandwiched by two other protein elements that contact the RNA. On one side, the long α1–β2 loop inserts into the loop-adjacent major groove and provides additional contacts with the RNA phosphate backbone. On the other side, the beginning of the C-terminal extension folds into a small α-helix bringing the downstream RGG repeats to contact the adjacent minor groove (Fig. 3h). Consequently, the interaction between RGG2 and the minor groove may direct the position of the second RNA-binding domain. Our solution structure supports that FUS RRM is the main binding site for the U1 snRNP, while the zinc-finger domain may interact with other RNA molecules.

**FUS contacts additional snRNAs during spliceosome assembly.** Besides the U1 snRNA, our FUS CLIP data also revealed strong enrichments for other snRNAs, which prompted us to further explore these interactions. Using cross-link analysis, we observed specific binding signatures in the 3′-stem loop (3′-SL) of U4 snRNA, a bulged loop (IL-1) in U5 snRNA and a linear stretch upstream of the GACAGA box in U6 snRNA (Supplementary Fig. 5a, b). Intriguingly, these cross-link sites are solvent exposed and clustered in the spliceosomal pre-B and B complexes, indicating that the interactions occur in the context of the spliceosome as opposed to individual snRNPs (Supplementary Fig. 5c, d). To address if these interactions are also mediated via the RRM domain, we incubated our $^{15}$N-labelled FUS-RRM construct with snRNA fragments encompassing U4 3′-ISL (nucleotides 93–109), U5 IL-1 (nucleotides 4–18 followed by a UUCG tetraloop and nucleotides 59–77) and U6 5′-UAUA-CUAA-3′ (nucleotides 21–28). Indeed, we found that all three RNAs induced chemical shift perturbations in the β-sheet surface and α1–β2 loop of the RRM as well as in RGG2, consistent with a direct interaction in vitro (Supplementary Fig. 5e). Overall, our results suggest that FUS employs its RRM to sequentially interact with multiple snRNAs as it escorts the spliceosome through the assembly phase of the splicing cycle.

**The FUS–U1 snRNA interaction is altered in the cytoplasm.** Consistent with cytoplasmic localisation of the ΔLC-FUS-P525L construct, we did not observe significant binding to intronic regions and lessened the enrichment of the predominantly nuclear snRNAs (Supplementary Fig. 6a, b). However, the U1 snRNA was still preferentially targeted among snRNAs (Supplementary Fig. 6c) and displayed an altered binding footprint: while one cross-link cluster confirmed the binding site of the RRM at SL3, we observed an additional cytoplasm-specific peak overlapping a GGU motif adjacent to the Sm site (Fig. 4a), with GGU being the preferred sequence bound by the zinc finger of FUS[9]. Hence, we decided to further study this interaction

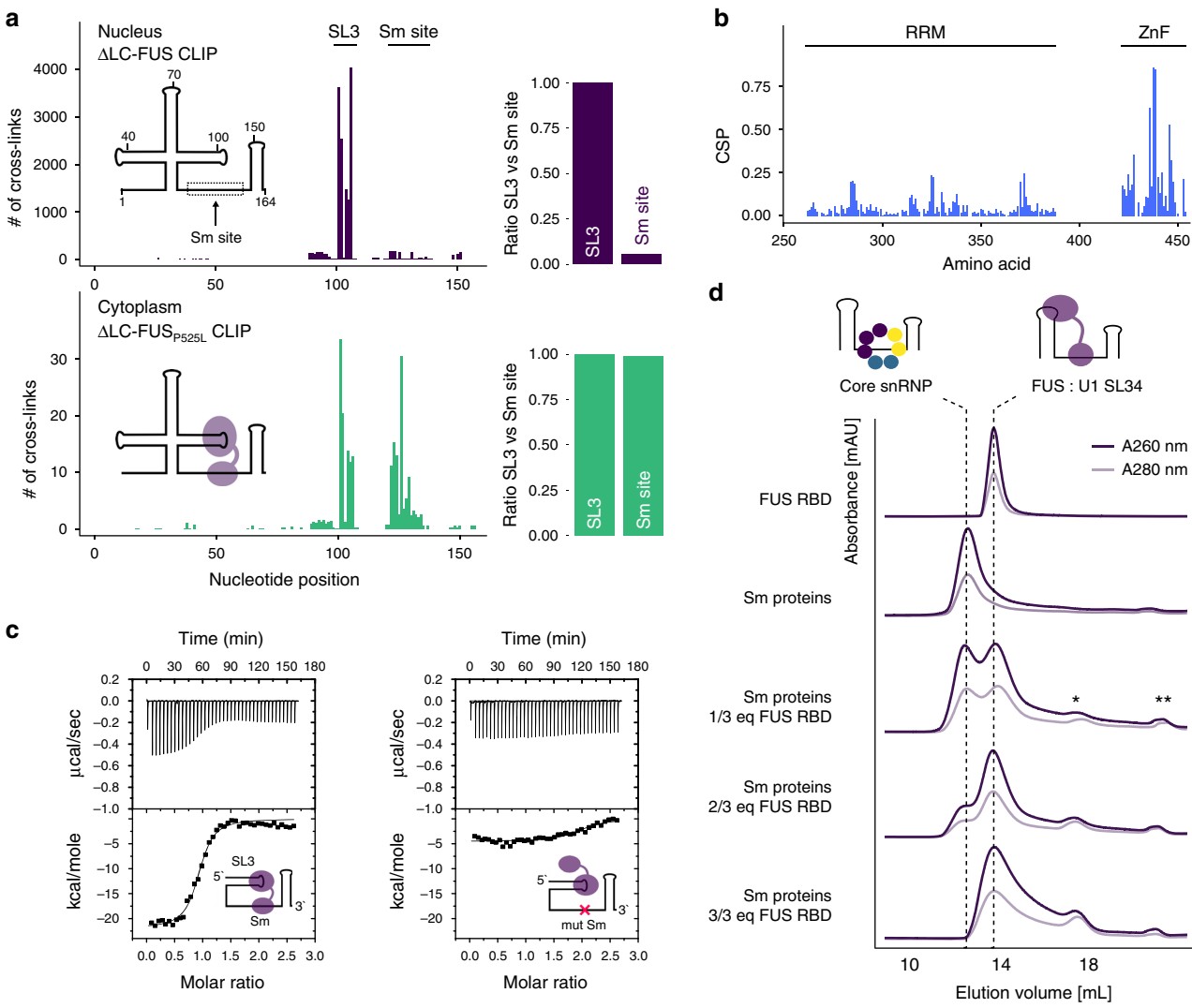

**Fig. 4 ALS-associated FUS cross-links to the Sm site of pre-U1 snRNA. a** Number of cross-links along U1 snRNA nucleotide sequence for the nuclear and cytoplasmic ΔLC-FUS constructs. The bar plots display the ratio of cross-links in the Sm site (nucleotides 123–135) relative to SL3 (nucleotides 99–110). **b** Bar plot showing the chemical shift differences in FUS–RBD upon addition of U1 SL34 RNA. $n = 3$. **c** ITC measurements of FUS with WT SL34 and mutSm SL34 RNA. $n = 3$. **d** Core snRNPs were assembled in vitro in the presence of the increasing amount of FUS. The resulting complexes were separated by analytical size-exclusion chromatography. Absorbance at 260 and 280 nm is shown. *unbound FUS–RBD, **unbound Sm proteins. $n = 3$.

in vitro using our recombinant FUS–RBD construct and a U1 snRNA fragment encompassing stem-loops 3 and 4 with the intervening Sm site (SL34). Using NMR spectroscopy, we found that addition of SL34 to uniformly [15]N-labelled FUS–RBD induced chemical shift perturbations of NMR resonances of the RRM as well as the zinc finger, indicating that both RNA-binding domains are involved in the interaction with the naked U1 SL34 RNA (Fig. 4b and Supplementary Fig. 6d). We then used isothermal titration calorimetry (Fig. 4c) and electrophoretic mobility shift assay (Supplementary Fig. 6e) to determine a binding affinity in the low nanomolar range ($K_d \sim 70$ nM). Notably, this interaction is sensitive to mutation of the GGU adjacent to the Sm site, confirming the cytoplasm-specific binding site identified by CLIP (Fig. 4c). Hence, we wondered if such a bipartite mode of RNA recognition would be compatible with the assembly of the heptameric Sm ring, which is an essential step during snRNP biogenesis that occurs in the cytoplasm[30]. To assess this, we performed core snRNP assembly assays in a cell-free environment by mixing SL34 with recombinant Sm proteins. Incubation of the RNA with either FUS or Sm proteins alone led to the formation of two distinct complexes that could be

separated by analytical size-exclusion chromatography (Fig. 4d). Intriguingly, titration of FUS to the Sm proteins impaired the formation of core snRNPs in a dose-dependent manner, confirming that the interaction between FUS and the U1 snRNA is incompatible with spontaneous core snRNP assembly in vitro. At equimolar amounts, FUS effectively outcompeted the Sm proteins to associate with the Sm site (Supplementary Fig. 6f). In mature U1 snRNPs, the Sm ring is stabilised by the N-terminal domain of U1-70K, which wraps around this core domain and could prevent nuclear FUS from displacing Sm proteins in the context of pre-mRNA splicing[27]. Consistent with this hypothesis, the presence of U1-70K (aa 1–216) reduces the capacity of FUS to impair core snRNP assembly by ~75% (Supplementary Fig. 6g).

**FUS and cellular stress impair snRNP biogenesis in ALS models.** To further explore the effects of FUS on snRNP biogenesis in a physiological cellular model, we then used CRISPR/Cas9 to target the endogenous *FUS* locus and generate isogenic hiPSC lines harbouring the ALS-associated P525L mutation as well as complete knockout of the *FUS* gene using the CRISPR-

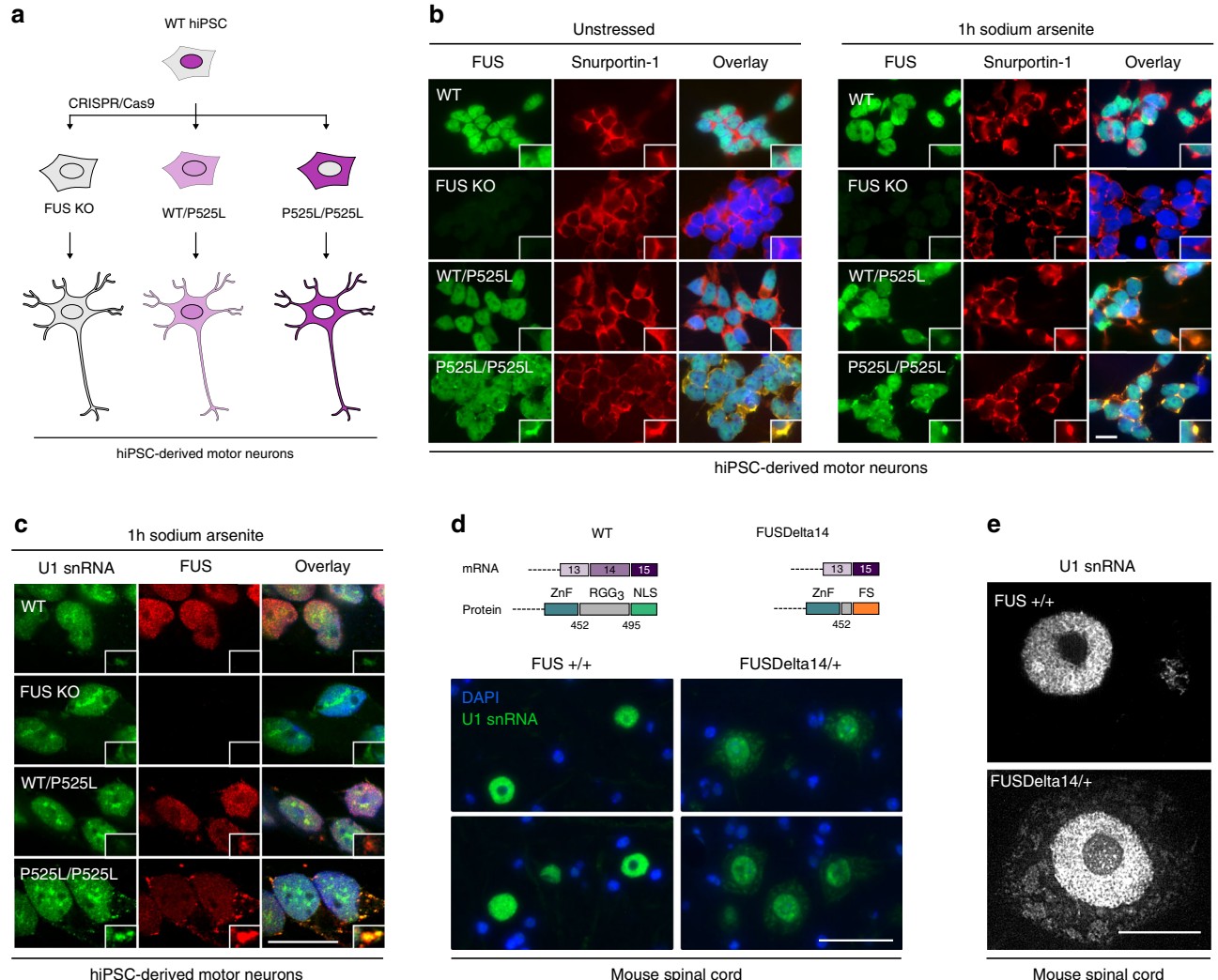

**Fig. 5 ALS-associated FUS traps U1 snRNA in the cytoplasm of motor neurons. a** Schematic representation of the genome-edited hiPSC-derived motor neurons used in this study. **b** Immunostaining in hiPSC-derived motor neurons with and without sodium arsenite treatment. FUS (green) and Snurportin-1 (red) were visualised using respective antibodies. Nuclei were counterstained with DAPI. Scale bar = 15 μm, n = 2. **c** Combined FISH and immunostaining in hiPSC-derived motor neurons following sodium arsenite treatment. FUS was visualised using anti-FUS antibodies (red), U1 snRNA with 6-FAM azide-labelled antisense probe (green). Nuclei were counterstained with DAPI. Scale bar = 15 μm, n = 1. **d** RNA FISH in 18-month spinal cord sections of WT and 'FUSDelta14' mice. U1 snRNA was visualised using 6-FAM azide-labelled antisense probe. Scale bar = 75 μm, n = 5. **e** High-resolution images of motor neurons from the same sections as the images in panel d. Scale bar = 15 μm, n = 5.

Trap approach[31] (Fig. 5a). Successful editing was verified by DNA sequencing, and the expression of pluripotency markers was confirmed by immunostaining (Supplementary Fig. 7a, b). To examine the cell-type predominantly affected in disease, we then differentiated our hiPSCs to motor neurons (MNs), employing a previously described protocol[32] (Supplementary Fig. 7c). In absence of stress, a small subset of homozygous MNs formed FUS condensates, and this behaviour was not observed for WT FUS-expressing MNs (Fig. 5b). However, most motor neurons did not display FUS inclusions, possibly due to their developmentally immature state[33]. We therefore induced cytoplasmic FUS condensation with sodium arsenite (SA) treatment for 1 h. The resulting FUS condensates stained positively for U1 snRNA (Fig. 5c) as well as the snRNP-specific import factor Snurportin-1 (Fig. 5b). The specificity of the FISH probe was verified by northern blotting using radiolabelled antisense probes (Supplementary Fig. 7d). This finding corroborates our recent analysis of the RNA content of purified FUS-containing droplets showing the presence of snRNA[23]. Nevertheless, we noted that SA-treatment also induced condensation of snRNP intermediates in the absence of FUS, suggesting that stress critically contributes to snRNP biogenesis defects in cellular FUS-linked ALS models. Indeed, Snurportin-1 condensates containing several snRNAs also formed in sodium arsenite treated hiPSCs and co-stained with the stress granule marker T-cell intracellular antigen-1 receptor (TIAR) (Supplementary Fig. 8a–e).

To circumvent the pitfalls of artificially added stressors and to study snRNP biogenesis in naturally aged tissue, we turned to an animal model and performed RNA fluorescence in situ hybridisation (FISH) in spinal cord sections of 18-month old 'FUSDelta14' mice. These mice harbour an ALS-associated splice site mutation that deletes the NLS and display a progressive loss of motor neurons in adult mice in the absence of FUS aggregation[16]. Strikingly, the FISH signal for U1 snRNA is clearly increased in the cytoplasm of spinal motor neurons of heterozygous 'FUSDelta14' mice compared to their wild-type littermates (Fig. 5d). In addition, we noted that U1 snRNA penetrates the nucleolus in 'FUSDelta14' mice (Fig. 5e), a feature

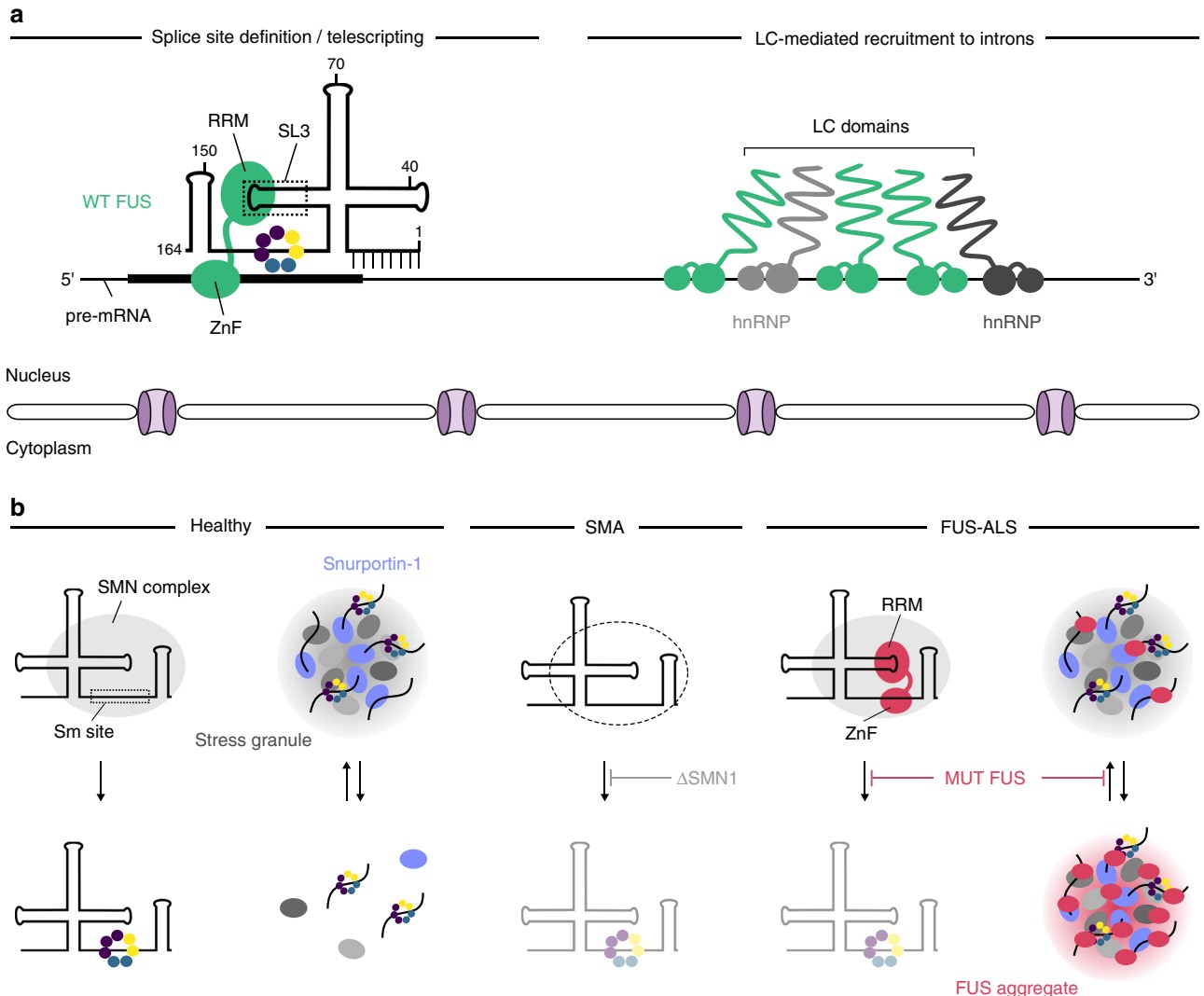

**Fig. 6 Bipartite interactions at the core of FUS' role in physiology and disease. a** In the nucleus, the RRM domain of FUS contacts the U1 snRNP at stem-loop 3, which positions the zinc-finger domain to interact with the pre-mRNA. Such intermolecular bipartite interactions could help to position the U1 snRNP on pre-mRNA to define 5'-splice sites and suppress premature polyadenylation. The LC domain of FUS allows FUS to interact with hnRNPs and promotes its recruitment to introns. **b** In the cytoplasm, U1 snRNPs undergo maturation with the SMN complex loading the Sm ring onto the snRNA. In SMA, this assembly activity is impaired due to insufficient amounts of the SMN protein. ALS-associated FUS engages in an aberrant intramolecular bipartite interaction with the U1 snRNA and interferes with core snRNP assembly in a toxic gain-of-function manner. Upon cellular stress, core snRNPs are sequestered into stress granules which may mature into the irreversible FUS aggregates observed in the disease.

that has been described for the Cajal body marker Coilin and Sm proteins under conditions of impaired snRNP biogenesis[34,35]. Together, these findings show that ALS-associated FUS traps a pool of U1 snRNA in the cytoplasm and disturbs biogenesis of U1 snRNP in vivo.

## Discussion

### Functional implication of the physiological interaction between FUS and U1 snRNP.
The spliceosome is a highly dynamic molecular machinery that assembles on each intron to catalyse its removal from nascent transcripts[36]. In the early stages of this process, the U1 snRNP binds to and defines the future 5'-splice site. Our CLIP data revealed that U1 snRNP is the preferred FUS partner in physiological conditions and the interaction occurs between FUS RRM and U1 snRNA stem-loop 3, which could have promoted the conservation of this unstructured loop. In contrast, stem-loop 4 of the U1 snRNA, which is also free from U1-specific proteins, harbours a UUCG tetraloop known to adopt

a structured conformation incompatible with FUS binding[37] and previously shown to be involved in the general mechanism of splicing through interaction with the U2 snRNP component SF3A1[38] or in alternative splicing via its interaction with the splicing factor polypyrimidine tract-binding protein 1[39]. Indeed, both stem-loops could act as hubs for splicing factors involved in the general mechanism of splicing or in its regulation. The solution structure of the FUS RRM bound to stem-loop 3 reveals that the RRM recognises the YNY motif in the 3'-part of the loop, while two positively charged lysines (K315/K316) in the α1–β2 loop contact the phosphate backbone in the major groove of the stem. These lysine residues have recently shown to be acetylated, which impairs the ability of FUS to bind RNA[40]. However, the structure further shows how the C-terminal RGG2 repeats wrap around the loop-adjacent minor groove to probably orient the other RNA-binding domain. Since we previously described how FUS achieves high-affinity RNA binding by combining both its RNA-binding domains, we propose that FUS could form a bridge

between U1 snRNP and other RNA molecules, such as the pre-mRNA or other spliceosomal RNAs (Fig. 6a). In agreement, the recent cryo-EM structure of the human pre-B spliceosomal complex revealed proximity between the stem-loop 3 and the exon of the pre-mRNA[28] and our structural simulations suggest that the RRM and the zinc finger of FUS could recognise RNA elements separated by up to 80 Å. Such intermolecular bipartite interactions could explain how FUS positions the U1 snRNP on pre-mRNA to modulate 5′-splice site selection and repress premature polyadenylation[41,42]. In a wider context, our work challenges the idea that splicing factors employ their RNA-binding domains to contact exclusively pre-mRNAs. Given that CLIP studies of numerous splicing factors focused on protein-coding RNAs, re-evaluation of published datasets with respect to snRNA binding could provide novel insights into the mechanisms of splicing regulation.

**Impaired snRNP assembly links FUS-ALS to SMA.** Disrupted snRNP biogenesis has been linked to motor neuron degeneration in spinal muscular atrophy (SMA), a childhood neuromuscular disorder caused by insufficient levels of survival motor neuron (SMN) protein, whose best-characterised role is to chaperone the assembly of small nuclear ribonucleoproteins (snRNPs)[30]. In particular, strong evidence comes from the findings that only SMN constructs that retain snRNP assembly activity are able to rescue SMA animal models and that injection of purified snRNPs rescues motor axon defects in a zebrafish model[43–45]. Impaired snRNP assembly induces downstream defects in RNA processing, which is thought to promote motor neuron death, though the molecular targets are still unclear. In addition, mutations in the snRNA genes *RNU4ATAC* and *RNU12* as well as *TOE1*, which encodes an exonuclease involved in snRNP biogenesis, cause distinct syndromes that share a strong neurological component[46–48]. Thus, the central nervous system appears to be particularly vulnerable to alterations in snRNP homeostasis. How exactly altered snRNP levels or profiles lead to disease remains unknown.

Several studies have explored the link between ALS and SMA. Genetic evidence suggests that abnormal *SMN1/2* gene copy numbers modulate the risk and severity of ALS humans whereas SMN overexpression delays motor neuron loss in SOD1(G93A) ALS mice[49–51]. Furthermore, ALS-associated FUS interacts with the SMN protein and sequesters it into cytoplasmic condensates while evoking a loss of SMN-containing nuclear structures called Gemini of Coiled Bodies (GEMs), a hallmark of SMA[15,25,52,53]. An additional link is provided by the transcriptional activator complex ASC-1, whose interaction with the RNA polymerase II machinery is disturbed by SMA-causing mutations in one of its components or ALS-causing mutations in FUS[54].

In this study, we used CLIP to identify an aberrant intramolecular bipartite interaction between ALS-associated FUS and the U1 snRNA, where the zinc-finger domain contacts a GGU motif in the Sm site and interferes with cytoplasmic U1 snRNP assembly in vitro and in a mouse model of FUS-linked ALS (Fig. 6b). Besides this toxic interaction, we propose that a second mechanism exacerbates the U1 snRNP biogenesis defect in a stress-dependent manner. In hiPSC-derived motor neurons, oxidative stress induced the condensation of core snRNPs into stress granules which co-localised with ALS-associated, but not wild-type FUS. Therefore, ALS-associated FUS could irreversibly sequester snRNP assembly intermediates by promoting the maturation of stress granules into pathological FUS aggregates (Fig. 6b). This is supported by evidence suggesting that stress granules may seed aggregate formation in ALS[10]. In addition to directly contacting cytoplasmic snRNAs, there is also evidence

that ALS-associated FUS aberrantly interacts with the assembly factor SMN and sequesters it into cytoplasmic condensates[25,52,53]. Thus, FUS could disturb snRNP biogenesis via different mechanisms.

Altogether, our results are consistent with the adult onset of the disease and provide a mechanistic explanation for the previously reported snRNP biogenesis defects in FUS-linked ALS[25,55–57]. The resulting alterations in snRNP homeostasis could explain how in mice ALS-associated FUS leads to mis-splicing of pre-mRNAs that are not regulated by nuclear FUS, but instead are sensitive to levels of the core splicing machinery, such as the SmB/B' pre-mRNA[15,58,59]. Subsequent work will be required to assess the impact of FUS on steady-state snRNP levels in disease-relevant cell types. Taking into consideration that disease onset is delayed by at least a decade in ALS compared to SMA, mild changes could already significantly contribute to the disease mechanism. Notably, dysregulated snRNP homeostasis has also been linked to TDP-43 and C9ORF72-linked ALS as well as the sporadic form of the disease[60–62]. Although the downstream effects and their impact on the disease remain to be investigated, it is important to note that not only pre-mRNA splicing but also polyadenylation is sensitive to reduced U1 snRNP levels[63]. In summary, our findings provide the molecular details of an RNA-based toxic gain-of-function of FUS in the cytoplasm causing a molecular defect that strengthens the link between FUS-linked ALS and SMA, with both motor neuron disorders displaying cytoplasmic snRNP biogenesis defects.

## Methods

**NMR spectroscopy.** All NMR spectroscopy measurements were performed using Bruker AVIII 600 MHz, AVIII 700 MHz and Avance 900 MHz spectrometers equipped with cryoprobes. The data were processed with Topspin 3.1 (Bruker) and analysed with CARA[64]. All the NMR experiments were performed in the NMR buffer that contains 10 mM sodium phosphate buffer pH 6.8, 50 mM NaCl, 2 mM DTT at 303 K (for FUS RRM-SL3) or 313 K (for FUS RDB-U1 snRNP/SL3/SL34).

**NMR titrations.** To monitor the interaction between FUS and U1 snRNP, a 20 μM solution of $^2$H, $^{15}$N, $^{13}$C ILV FUS–RDB was titrated with U1 snRNP at 313 K. At each titration step, a 2D $^1$H-$^{13}$C HMQC spectrum as well as a 2D $^1$H-$^{15}$N TROSY spectrum were recorded. A similar procedure was followed when the $^2$H, $^{15}$N, $^{13}$C ILV FUS–RDB were titrated with U1 SL3. To monitor the interaction between FUS–RDB and U1 SL34, a 100 μM solution of $^{15}$N-labelled FUS–RDB was titrated with U1 SL34 at 313 K. At each titration step, a 2D $^1$H-$^{15}$N TROSY spectrum was recorded. The NMR titration of FUS RRM with U1 SL3 was performed by adding unlabelled aliquots of U1 SL3 into a 100 μM solution of $^{15}$N-labelled FUS RRM. At each titration step, a 2D $^1$H-$^{15}$N TROSY spectrum was recorded. The NMR titration of U1 SL3 with FUS RRM was performed by adding unlabelled aliquots of FUS RRM into a 100 μM solution of $^{13}$C-labelled U1 SL3. The formation of the complex was monitored by recording a couple of 2D $^1$H-$^{13}$C HSQC spectra (centred on aliphatic or aromatic regions of RNA) after each titration step.

**Resonance assignment.** The resonance assignment of the FUS RRM construct (aa 260–390) in complex with U1 snRNA SL3 was performed by analysing classical triple-resonance experiments (3D HNCO, 3D HNCACB and 3D CBCA(CO)NH). Side-chain assignment was performed based on 3D (H)C(CO)NH, 3D H(C)(CO) NH recorded in H$_2$0 and 3D H(C)CH-TOCSY and 3D (H)CCH-TOCSY recorded in 100% D$_2$0. The resonance assignment of stem-loop 3 was performed by combining homonuclear experiments recorded in H$_2$O or 100% D$_2$O (2D $^1$H–$^1$H TOCSY and 2D $^1$H–$^1$H NOESY) and heteronuclear NMR spectroscopy experiments recorded with a $^{13}$C-labelled sample of U1 SL3 in 100% D$_2$O (3D $^1$H–$^{13}$C HSQC NOESY and 3D H(C)CH-TOCSY). RNA base pairing was deduced from cross-strand NOEs in RNA helical regions. Sugar puckers were identified by analysing the 2D $^1$H–$^1$H TOCSY (Tm=25 ms) and syn or anti conformations were deduced from NOE patterns of H6 and H8 resonances. The resonance assignment of the free RNA stem loop was then transferred to the bound RNA stem loop. Intermolecular NOEs were identified in 2D F2 $^{13}$C-filtered NOESY and 3D $^{13}$C-(F1-edited, F3-filtered) NOESY HSQC spectra recorded with $^{15}$N–$^{13}$C-labelled protein and unlabelled RNA or with $^{15}$N–$^{13}$C-labelled RNA and unlabelled protein in 100% D$_2$O with a mixing time of 120 ms.

**Structure calculations.** The resonance assignment of the bound protein was used as input for automatic peak picking and NOESY assignment using ATNOS-CANDID[65]. Resulting peak lists were checked and supplemented manually. RNA

and intermolecular NOESY peaks were manually assigned and calibrated. Protein peaks were then re-assigned with the NOEASSIGN module of CYANA3.96[66] and manually checked. Structure calculations were performed using a list of unambiguous intramolecular NOE-derived distances for the protein and the RNA, unambiguous intramolecular NOE-derived distances and ambiguous restraints for the C-terminal RGG tail using CYANA. Due to the strong overlap of arginine and glycine side-chain resonances of the C-terminal RGG tail, intermolecular NOEs were treated using ambiguous restraints. Successive calculations allowed us to progressively remove the most violated ambiguous restraints before cartesian refinement. In addition, protein hydrogen bonds in secondary structures as well as dihedral angles restraints of the protein backbone were derived from the analysis of the backbone chemical shifts using TALOS + were also included. Finally, RNA base pairing and loose sugar pucker restraints were applied to constraint the double-stranded part of the RNA. Final calculations were performed using CYANA and out of 500 structures generated, the 50 structures with the lowest target function were further refined in cartesian space with the SANDER module of AMBER14[67] using ff14SB force field. Lowest energy models were then selected.

**Analytic size-exclusion experiments**. Analytic size-exclusion chromatography experiments were performed using Superdex 200 10/300 GL in 10 mM sodium phosphate buffer pH 6.8, 100 mM NaCl and 2 mM DTT. For the formation of the Sm core, each Sm protein was added in a test tube at a final concentration of 20 μM together with 5 μl of RNAsin (Invitrogen) and incubated at 37 °C. After 5 min at 37 °C, the RNA was added, incubated for another 5 min at 37 °C, the sample volume was then adjusted to 250 μl and directly load on the size-exclusion column (S200 increase, GE Healthcare). A similar procedure was applied to prepare the FUS–RDB–SL34 complex. For the competition between FUS and Sm protein for SL34 binding, constant amounts of Sm proteins were incubated with various amount of FUS–RBD before the addition of the RNA. To test the effect of U1-70K on the competition between FUS and Sm protein for SL34 binding, we incubated together the equimolar amount of Sm proteins, FUS RBD and U1-70K (1–216) before the addition of the RNA in the solution. Data were directly integrated using Unicorn (GE Healthcare) and analysed using GraphPad.

**Isothermal titration calorimetry**. ITC experiments were performed on a VP-ITC microcalorimeter (Microcal). All proteins and RNA were extensively dialysed in 10 mM sodium phosphate buffer pH 6.8, 50 mM NaCl and 1 mM β-mercaptoethanol. For the titration between FUS RRM and SL3, the protein was concentrated to 600 μM and the RNA to 50 μM. For the titration between FUS–RBD and SL34 (or SL34mut), the protein was concentrated to 60 μM and the RNA to 6 μM. The titrations were performed at 25 °C using a single injection of 2 μL followed by 6 μL injection every 300 s with a stirring rate of 307 rpm. Raw data were integrated and analysed using Origin 7.0 using a 1:1 stoichiometry. For the titration of FUS RRM and SL3, $n = 0.85 \pm 0.02$, $K = (1.25 \pm 0.08) \times 10^5$ M, $\Delta H = (-2.09 \pm 0.08) \times 10^4$ kcal/mol and $\Delta S = -45.8$ kcal/mol. For the titration of FUS–RBD and SL34, $n = 0.92 \pm 0.01$, $K = (1.37 \pm 0.23) \times 10^7$ M, $\Delta H = (-2.13 \pm 0.04) \times 10^4$ kcal/mol and $\Delta S = -38.9$ kcal/mol.

**CLIP-Seq**. SH-SY5Y cell lines were grown to 80% confluency in two 15-cm dishes per biological replicate. The cells were washed once with ice-cold PBS, covered with 10 ml of PBS and cross-linked at 254 nm and 150 N/cm² using a Bio-Link® crosslinker (Vilber Lourmat, BLX-E). Subsequently, the cells were scraped off the plates and spun down at $500 \times g$ and 4 °C for 5 min. After removal of the supernatant, the pellets were shock-frozen in liquid nitrogen and stored at −80 °C until use. Cells were lysed in 2 ml RIPA buffer (Thermo Scientific) supplemented with 2× HALT protease inhibitor [Pierce] and 0.5 U/μl RNase inhibitor (Lucigen) and incubated on ice for 15 min. To mask-free biotin and biotinylated proteins, 1 U avidin (Novex) was added to the lysate. Then, cellular debris was removed by centrifugation at $15,000 \times g$ and 4 °C for 15 min. The cleared lysates were incubated with 15 μL RNaseI (Thermo Scientific) dilution (1:250 in RIPA buffer) and 7.5 μL Turbo DNase (Ambion) at 37 °C for 7.5 min and then cooled on ice for 3 min to partially digest cross-linked RNAs. Per IP, 100 μL of MagStrep Type 3 XT beads (IBA lifesciences) were washed twice with IP wash buffer (50 mM HEPES–NaOH pH 7.3, 300 mM KCl, 0.05 % NP-40). Subsequently, the protein–RNA complexes were bound to the beads head over tail for 1.5 h at 4 °C. After four washes with IP wash buffer, the beads were resuspended in 150 μL dephosphorylation mix (8 U antarctic phosphatase enzyme [NEB] in 1× reaction buffer and 0.5 U/μL NxGen RNase inhibitor) and incubated at 37 °C and 900 rpm for 20 min. After that, the beads were washed twice with IP wash buffer and twice with PNK buffer (50 mM Tris-HCl pH 7.6, 150 mM NaCl, 1% NP-40, 1% sodium deoxycholate, 0.1% SDS). Then, 150 μL 5'-phosphorylation mix (100 U T4 PNK (Thermo Scientific), 200 μCi γ-³²P-ATP (3000 Ci/mmol) (Hartmann Analytics), 3 mM ATP in 1× reaction buffer A and 0.5 U/μL NxGen RNase inhibitor) were added, and the samples were incubated at 37 °C and 900 rpm for 45 min. Finally, the beads were washed four times with PNK buffer. Protein–RNA complexes were eluted by heating in 45 μL 2× LDS sample buffer (supplemented with 5 mM biotin) at 70 °C for 10 min, separated on a NuPage 4–12% Bis-Tris gradient gel in MOPS buffer and transferred to a nitrocellulose membrane (Life Technologies). Paper arrows were dipped in the last PNK wash to mark 40 kDa and 80 kDa bands on the dried membrane,

which was subsequently wrapped in cling foil and exposed to a phosphorimager screen for 38 h. The membrane sections containing the desired protein–RNA complexes were excised using a clean blade, transferred to Eppendorf tubes and grinded using a pipette tip. Next, the membrane fragments were incubated with 400 μL proteinase K mix (20% proteinase K (Invitrogen) diluted in water) at 37 °C for 30 min. Following the addition of an additional 200 μL proteinase K mix, the reactions were incubated again for 30 min. RNA was isolated from the supernatant by addition of 1 volume acidic phenol:chloroform:isoamylalcohol (25:24:1) and centrifugation at $13,000 \times g$ for 10 min. The aqueous phases (350 μL) were transferred to fresh tubes and precipitated upon addition of 35 μL volume 3 M NaOAc pH 4.6, 2 μl glycoblue (Ambion) and 1 mL absolute EtOH at −80 °C for 30 min. Subsequently, the precipitated RNAs were spun down at $16,000 \times g$ and 4 °C for 30 min, washed with 75% EtOH and resuspended in 20 μL DEPC water. Library preparation of the CLIP samples was performed at Fasteris SA. Following adapter ligation using the TruSeq small RNA sample preparation kit (Illumina) and 25 cycles of PCR amplification, the libraries were sequenced on the HiSeq 2500 platform (Illumina) using $1 \times 125$ bp single-end (ΔLC-FUS) and $2 \times 125$ paired-end (FUS and ΔLC-FUS$_{P525L}$) cycles. For input RNA sequencing, 2 μg of the total RNA from the cell lines were ribo-depleted using RiboCop (Lexogen) according to the manufacturer's instructions. Library preparation was performed using the TruSeq stranded mRNA library preparation kit (Illumina). Samples were sequenced on a HiSeq 2500 platform using 150 cycles in paired-end (ΔLC-FUS) and single-end mode (FUS and ΔLC-FUS$_{P525L}$). CASAVA (v1.8.2) (Illumina) was used to convert Bcl files to FASTQ format.

**RNA Immunoprecipitation (RIP)**. In total, $1.5 \times 10^7$ HeLa TS-FUS (1.0) (described in ref. [68]) and HeLa TS-mRBD-FUS (1.0) cells were lysed in 2 mL gentle hypotonic lysis buffer (10 mM Tris pH 7.2, 10 mM NaCl, 2 mM EDTA, 0.5% Triton-X-100) supplemented with 2× HALT protease inhibitor (Pierce), 0.5 U/μL RiboLock RNase inhibitor (Thermo Scientific) and 2 U/mL Turbo DNase (Ambion). After 10 min incubation on ice, the NaCl concentration was adjusted to 150 mM, followed by addition of 1 U avidin (Novex) and another 5 min incubation on ice. The lysate was cleared by centrifugation at $16,100 \times g$ for 15 min at 4 °C. In total, 200 μL (~$1.5 \times 10^6$ cell eq.) and 50 μL (~$3.8 \times 10^5$ cell eq.) were put aside to serve as RNA and protein input, respectively. In all, 1.5 mL (~$1.1 \times 10^7$ cell eq.) were transferred to 100 μL MagStrep Type 3 XT beads (IBA lifesciences) equilibrated in HEPES-NET-2 buffer (50 mM HEPES pH 7.3, 150 mM NaCl, 0.1% Triton-X-100) and incubated head over tail at 4 °C for 1.5 h. After five washes with HEPES-NET-2 buffer, 2× LDS sample buffer was added to 1/5 of the beads (~$2.2 \times 10^6$ cell eq.) while 1 ml TriReagent (Invitrogen) supplemented with 0.14 M β-mercaptoethanol (AppliChem) was directly added to the remaining beads ($0.9 \times 10^7$ cell eq.). RNA was isolated according to the manufacturer's manual (Life Technologies), except that precipitation was performed overnight at −20 °C in the presence of 2 μL of GlycoBlue (Ambion). Following centrifugation for at $16,100 \times g$ for 1 h, and two subsequent washes with 80% EtOH, the pellets were dried and resuspended in DEPC-H₂O. In total, 2 μg of RNA were reverse-transcribed at 37 °C in 50 μL containing 1× small RNA RT buffer (10 mM Tris pH 8.0, 75 mM potassium, 10 mM dithiothreitol, 70 mM magnesium chloride, 0.8 mM anchored universal RT primer), 2 U/μL of RiboLock (Thermo Scientific), 10 mM dNTPs, 2.5 mM rATP supplemented with 5 U of *Escherichia coli* poly(A) polymerase (New England Biolabs) and 1 μL AffinityScript multiple temperature reverse transcriptase (Agilent). Reactions were heat-inactivated for 10 min at 85 °C. Reverse-transcribed material corresponding to 18 ng or 360 pg of RNA was amplified with Takyon qPCR Master Mix blue dTTP for SYBR (Eurogentec) and the corresponding primers (600 nM each) in a total volume of 20 μL using a Rotor-Gene 6000 cycler (Corbett). Primer sequences are listed in Supplementary Table 1.

**Fluorescence in situ hybridisation (FISH)/immunofluorescence in cell lines**. Immunofluorescence was performed before FISH. Cells were fixed with 4% PFA for 15 min and then washed twice with PBS and twice with 70 °C ethanol before permeabilization in 70% ethanol for 48 h at 4 °C. After washing three times for 5 min at RT with PBS, the slides were blocked three times for 10 min at RT with blocking buffer (1% BSA in PBS supplemented with 2 mM ribonucleoside vanadyl complex (RVC) (Sigma)). Primary antibodies were diluted in blocking buffer and incubated for 1 h at 37 °C and 1 h at room temperature. After three 5-min washes with blocking buffer, the secondary antibodies were added for 45 min at room temperature. The cells were then washed three times with PBS and post-fixed with 4% PFA for 5 min at room temperature to cross-link antibodies with their targets. Then, the slides were washed twice with 2× SSC (300 mM NaCl, 30 mM sodium citrate pH 7.0) and incubated with pre-hybridisation buffer (15% formamide, 10 mM sodium phosphate, 2 mM RVC in 2x SSC, pH 7.0) for 10 min at room temperature. Antisense probes were diluted to 0.5 ng/μl in hybridisation buffer (15% formamide, 10 mM sodium phosphate, 10% dextran sulfate, 0.2% BSA, 0.5 μg/μl *Escherichia coli* tRNA, 0.5 μg/μl salmon sperm DNA, 2 mM RVC in 2× SSC, pH 7.0) and hybridised to the cells overnight at 42 °C. The next day, the cells were subsequently washed (all wash steps at 42 °C) two times for 30 min with pre-hybridisation buffer and three times for 10 min in high-stringency wash solution (20% formamide, 2 mM RVC in 0.05× SSC, pH 7.0). After three washes in 2× SSC, the slides were mounted with aqueous vectashield mounting medium containing DAPI (Vectorlabs). Antibodies are listed in Supplementary Table 2.

**Fluorescence in situ hybridisation (FISH) in tissue**. Mouse spinal cord tissue was harvested as previously reported[16] according to applicable international, national, and institutional guidelines, including ARRIVE guidelines, for the care and use of animals and according to UK home office regulations. FISH was performed before Immunofluorescence. To dewax sections, the slides were incubated in xylene three times for 5 min. Then the sections were hydrated in a stepwise manner by incubations in 100% EtOH (2 × 2 min), 90% EtOH (1 × 2 min), 80% EtOH (1 × 2 min) and 70% EtOH (1 × 2 min) and finally distilled water (1 × 5 min). To retrieve antigens, the slides were boiled in 1 L citrate buffer (100 mM citrate pH 6.0) for 20 min in a microwave on high power. The slides were cooled by exchanging the buffer with slowly running cold tap water. Then they were transferred in distilled water (3 × 5 min) and hydrophobic barriers were created using a barrier pen. After one wash in 2 × SSC (1 × 5 min) the slide was incubated with pre-hybridisation buffer (15% formamide, 10 mM sodium phosphate, 2 mM RVC in 2× SSC, pH 7.0) at 42 °C for 30 min. The labelled antisense U1 probe was diluted to 500 pg/μL in hybridisation buffer (15% formamide, 10 mM sodium phosphate, 10% dextran sulfate, 0.2% BSA, 0.5 μg/μl *Escherichia coli* tRNA, 0.5 μg/μl salmon sperm DNA, 2 mM RVC in 2× SSC, pH 7.0) and incubated with the sections overnight at 42 °C. The next day, the slides were first washed at 42 °C 6 × 15min with high-stringency wash buffer (10–50% formamide, 0.05x SSC, 2 mM RVC) followed by washes in 0.05× SSC (3 × 10 min, 42 °C) and PBS (1 × 5 min, RT). Subsequently, the slides were incubated in Sudan Black for 5 min to reduce tissue auto-fluorescence. Following short incubation in PBS to remove the bulk of residual Sudan Black and four additional 5-min washes in PBS, the slides were mounted with aqueous Vectashield mounting medium containing DAPI (Vectorlabs).

**Image acquisition**. Images of SH-SY5Y cells were obtained with a non-confocal fluorescence microscope (Leica DMI6000 B) using a 60×/1.4 NA oil immersion lens and the LAS X software platform (Leica). Images of hiPSCs and hiPSC-derived motor neurons were taken with a non-confocal Eclipse Ti-2 epifluorescence microscope (Nikon) using the NIS-Elements AR software (Ver 5.01) and either a 20×/dry or 60×/1.4 NA oil immersion lens. Confocal images of hiPSC-derived motor neurons and mouse spinal cord were obtained with a super-resolution VT-iSIM microscope (Nikon) using a 100×/1.49 NA oil immersion lens. Deconvolution was performed with the NIS-Elements AR software (Ver 5.01) using the Richardson/Lucy algorithm with 20 iterations. For printing, brightness and contrast of individual channels were linearly enhanced using the Fiji software[69].

**Electrophoretic mobility shift assay (EMSA)**. To refold the RNA, SL34 RNA was first diluted to 250 pM in 1× binding buffer (10 mM HEPES pH 7.3, 100 mM KCl, 5 mM MgCl₂, 10 μg/ml yeast tRNA, 10 μg/ml salmon sperm DNA), incubated at 95 °C for 1 min and then at 65 °C for 1 min before allowing to cool down slowly to room temperature. For the binding reactions, 2 fmol RNA (100 pM concentration) were mixed with increasing concentrations of the FUS–RBD constructs (up to 2 μM) in 1× binding buffer for 1 h at room temperature. Subsequently, RNA gel loading buffer (5% glycerol, traces of bromophenol blue and xylene cyanol) was added and the protein–RNA complexes were separated on a non-denaturing 0.5× TBE 6% polyacrylamide gel in 0.5× TBE buffer under constant cooling. The gel was then fixed with EMSA fixing solution (5% glycerol, 12% methanol, 10% acetic acid), vacuum dried and exposed to a phosphorimager screen overnight.

**Genome editing**. Exon 15 of the *FUS* gene was targeted to introduce the P525L mutation using the pCRISPR-EF1a-SpCas9-P525 plasmid coding for the sgRNA targeting the sequence 5′-GGAGCCAGGCTAATTAATACGG-3′ using the strategy described in[70]. One day before transfection, 10 μM rock inhibitor Y-27632 (Stemcell Technologies) and 2 μM pyrintegrin (Stem cell technologies) were added to the stem cells. On the day of transfection, six wells of a six-well plate containing each 90% confluent stem cells in mTeSR1 containing rock inhibitor and pyrintegrin were transfected using TransIT®-LT1 Transfection Reagent (Mirus) according to the manufacturer's instructions. Here, each six-well was transfected with a total amount of 5 μg of DNA, transfecting 200 ng of pRR-Puro-P525 and 4.8 μg of a mix of pCRISPR-EF1a-SpCas9-P525 and the P525L donor plasmid for HDR. For each six-well, a different molar ratio of pCRISPR-EF1a-SpCas9-P525 and donor plasmid was used (1:1, 1:3, 1:6, 4:1, 3:1, 2:1). The day after transfection, the medium was changed to mTeSR1 containing 10 μM rock inhibitor and 2 μM pyrintegrin supplemented with the 5 μl of the HDR-enhancer L755507 (Sigma). Two days after transfection, cells were detached using Accutase (Thermo Fisher) and pooled on one 15-cm plate in mTeSR1 containing 10 μM rock inhibitor and 2 μM pyrintegrin supplemented with 0.5 μg/ml puromycin. The selection was maintained for one more day and rock inhibitor and pyrintegrin were maintained for 4 more days. Thereafter, colonies growing from single cells were picked, and gDNA was isolated for clone screening using TRIzol according to the manufacturer's instructions. The P525L genomic locus was amplified from the genomic DNA using the KAPA Taq ReadyMix PCR Kit according to the manufacturer's instructions. PCR products were purified over a preparative agarose gel using the Wizard SV Gel and PCR Clean-Up System (Promega). Purified PCR products were sequenced at Microsynth AG.

**Motor neuron differentiation**. Motor neurons differentiation was performed using a protocol based on[71] with modifications from[72], as previously described in ref. [32]. On day 1 of the differentiation, the hiPSCs were dissociated with TrypLE Express (Gibco, 12604021) and resuspended in DMEM/F12 + N2 & neurobasal + B27 (all Gibco, 31331028, 17502048, 21103049, 17504044) in a 1:1 ratio supplemented with 1× penicillin–streptomycin (Gibco, 15140122), 5 μM Y-27632 (Tocris, 1254), 40 μM SB431542 (Tocris, 1614), 200 nM LDN193189 (Tocris, 6053), 3 μM CHIR99021 (Tocris, 4423) and 200 μM ascorbic acid (Sigma-Aldrich, A4403). The cell suspension was then distributed in 96-well round-bottom plates (Thermo Scientific, 168136) with roughly 100,000 cells per well and spun down with 530 × g for 4 min to form embryonic bodies. On day 2, the embryonic bodies were transferred into six-well plates (Thermo Scientific, 140675), the medium was exchanged with a medium of the same composition as day 1. The medium was exchanged daily from days 3 to 9 with DMEM/F12 + N2 & neurobasal + B27 in a 1:1 ratio supplemented with 1× penicillin–streptomycin, 200 nM retinoic acid (Sigma-Aldrich, R2625), 500 nM SAG (Tocris, 4366) and 200 μM ascorbic acid. On day 10, the embryonic bodies were disassociated with Tryple Express and seeded in 12-chamber microscope slides (ibidi, 81201) and 96-well black/clear bottom plates (Corning, 353219) coated consecutively three times with 0.001% poly-L-ornithine (Sigma-Aldrich, P4957) and one time with 2 μg/ml laminin (Sigma-Aldrich, L2020) and 2 μg/ml fibronectin (Sigma-Aldrich, F2006) with each coating step being carried out at 37 °C overnight. The cells were seeded in a concentration of 200,000 cells per cm² in neurobasal + B27 supplemented with 1× penicillin–streptomycin, 5 μM Y-27632, 200 nM retinoic acid, 500 nM SAG, 200 μM ascorbic acid, 10 μM DAPT (Tocris, 2634), 10 ng/μl GDNF (R&D Systems, 212-GD) and 10 ng/μl BDNF (R&D Systems, 248-BDB). The medium was exchanged on days 11 and 12 with the same composition as day 10 albeit without Y-27632. The medium was exchanged on day 13 with the same composition as day 10 albeit without Y-27632 and DAPT. On day 14, the motor neurons were fixed with 4% PFA for 20 min at RT.

**RNA-seq data analysis**. Mapping of raw reads obtained from RNA-seq experiments was accomplished using STAR version 2.5.2a[73] with the parameters of the RNA-seq pipeline for long RNAs provided by the ENCODE Data Coordinating Center and the full ENSEMBL gene annotation version 90 of genome assembly GRCh38.

**CLIP-Seq data analysis**. Preprocessing: single- and paired-end samples were subjected to 3′ adapter trimming using cutadapt version 1.14[74]. Mapping and additional processing steps: the trimmed reads were mapped using STAR version 2.5.3a[73] with the same parameters as for RNA-seq samples. Putative PCR duplicates were filtered from both single- and paired-end libraries by applying samtools (version 1.8.3) utilities fixmate and markdup[75]. The intersection of alignments and gene annotation: To infer the location of the aligned reads with respect to specific gene annotation features (exons, introns, etc.), a filtered gene annotation was used. The filtered set only contained entries of genes annotated with support level 1 (all splice junctions supported by at least one trusted mRNA sequence) plus the following classes of non-coding RNAs: snRNA, snoRNA, scaRNA, scRNA, miRNA and lincRNA. If an alignment intersected with multiple annotated features, each feature was counted partially, with a weight proportional to the width of the intersection. The intersections of features with multi-mapping reads were further weighted with 1/# mappings of the read. Highly reproducible binding sites: For each sample, a set of highly reproducible FUS–RNA interaction sites was inferred by exploiting the deletions introduced by the reverse transcriptase during cDNA library preparation. First, only deletions not already annotated as SNPs in the ENSEMBL vcf file of gene annotation GRCh38 version 90 were considered. In regions of the alignment where the forward and reverse read overlapped, deletions were required to be identified in the alignment of both reads. Finally, only those deletion sites where the mutation frequency among all alignments overlapping this site was below 50% were retained. Clustering of deletion sites: individual deletion sites were combined to deletion regions if they were less than 11 nucleotides apart. Enrichment analysis: for the inferred deletion regions (see above), an enrichment ratio of alignments from the CLIP experiment and from a matching RNA-seq experiment was calculated as follows. Each deletion region was extended up- and downstream by 100 nucleotides. For CLIP and RNA-Seq samples, the raw numbers of single-end or read-pair alignments with at least one matching nucleotide in the defined region were identified and a pseudo-count of one was added for both values. Multi-mapping reads were counted towards each of the matching loci as 1/ number of matching loci.

The enrichment score was calculated as the ratio of library size normalised CLIP reads and library sized normalised RNA-seq reads that mapped to the region surrounding the site. The enrichment score was subsequently used as a metric to sort deletion sites, e.g., to compute the overlap of replicates with respect to transcripts with the highest enrichment scores. Secondary structure analysis: utilising the intersection of inferred binding sites from all three replicates of the full-length construct, secondary structure prediction was done with the RNAfold program of the ViennaRNA Package version 2.4.8[76]. For each condition, a foreground set of binding sites was selected as the intersection of highly reproducible sites from all replicates. A background set was obtained by shuffling the foreground sequences while preserving the dinucleotide frequencies with a tool from the meme suite version 4.12.0[77]. After obtaining secondary structure

predictions for each sequence through RNAfold, a position-wise mean base-pairing score is inferred for both sets independently assuming the following fold propensities: A: 0.365, C: 0.516, G: 0.663, U: 0.494. The final fold propensity score for each position was calculated as the ratio of the pairing score from the foreground and the pairing score from the background for this position. Nucleotide composition analysis: The same sequences as for the secondary structure analysis was used to calculate the nucleotide frequencies at given positions and plot them as nucleotide composition. The entire data analysis process was executed as a workflow which was created with snakemake version 4.3.0[78]

**Statistics and reproducibility**. To satisfy the requirements for standard statistical tests and to ensure the robustness of our results, all experiments yielding quantitative data were performed in triplicate, which is the standard for molecular biology experiments. Because of the non-linear transformation of Ct values in relative quantifications using the $2^{\Delta\Delta CT}$ method, we performed statistical analyses using log-transformed values and employed two-sided Welch's $t$ test considering the unequal variances observed between conditions.

**Reporting summary**. Further information on research design is available in the Nature Research Reporting Summary linked to this article.

## Data availability
The accession number for the FUS-RRM:SL3 structure reported in this paper is PDB: 6SNJ. The accession number for FUS-RRM:SL3 chemical shifts reported in this paper is BMRB:34427. Input total RNA-Seq data and high-confidence FUS-binding sites inferred from CLIP were uploaded to GEO with the accession number GSE139263. All data supporting the findings of this study are available from the corresponding author upon reasonable request.

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

## Acknowledgements

This research and related results were made possible through the support of the NOMIS Foundation (M.-D.R.), the UK Dementia Research Institute which receives its funding from UK DRI Ltd, funded by the UK Medical Research Council, Alzheimer's Society and Alzheimer's Research UK (M.-D.R.), the John and Lucille van Geest foundation (M.-D.R.), the Swiss Life Jubiläumsstiftung (M.-D.R.), the Motor Neurone Disease Association (A.D.; 867–791), the NCCR RNA and Disease funded by the Swiss National Science Foundation (F.H.-T.A., M.Z.), and the Swedish Medical Research Council (Vetenskapsrådet, 2016–02112) (E.H.). We are grateful to Muriel Fragnière and Tosso Leeb of the NGS platform at the University of Bern, and George Chenell of the Wohl Cellular Imaging Centre at King's College London (including the Nikon Ti-E eclipse microscope funded by Alzheimer's Research UK; ARUK-EG2013B-1) for technical support. We thank Rahel Kräuchi (Department of Chemistry and Biochemistry, University of Bern) for the stable HeLa TS-FUS cell lines, Jacqueline C. Mitchell (King's College London) for valuable discussions, Nicole Kleinschmidt & Karin Schranz (Department of Chemistry and Biochemistry, University of Bern) and Joana Conde de Almeida de Sousa Furtado & Tanisha Lewis (UK DRI at King's College London) for their technical and organisational assistance.

## Author contributions

Conceptualisation: D.J., S.C., F.H.-T.A. and M.-D.R.; experiments: D.J., S.C., S.R., J.M. and C.S.; data analysis: R.S., F.G., M.C., D.J. and S.C.; resources: A.D., F.E.L. and C.V.S.; writing—original draft: D.J. and S.C.; writing—review and editing: D.J., S.C., R.S., C.S., M.C., A.D., F.E.L., E.H., M.Z., F.H.-T.A. and M.-D.R.; funding and supervision: E.H., M.Z., F.H.-T.A. and M.-D.R.

## Competing interests

The authors declare no competing interests.
