## [Peer Review File · Nature Communications]

REVIEWER COMMENTS

Reviewer #1 (Remarks to the Author):

Please see attached report.

The manuscript “Aberrant interactions of FUS...” NCOMMS-20-23834, is a real tour-de-force combining several different cutting edge techniques. In my opinion, it certainly meets the Nat. Comm. journal requirements of being technically sound, presenting strong evidence for the conclusions and presenting novel findings which stimulate new directions in the field. In particular, the solution structure of the FUS RBD + U1 snRNP SL3 RNA complex and cell fluorescence microscopy studies make a strong case that splicing factors like FUS and TDP-43 do not just bind mRNAs but can bind to other nuclear RNAs and aberrantly transport them outside nucleus.

Minor comments:

1. Abstract and Discussion (page 13). In my opinion, the connection between spinal muscular atrophy / survival of motor neuron protein and ALS / FUS + U1 snRNA is not so clear as the authors do not seem to have reported any results on the survival of motor neuron protein. Therefore, it seems that the “reinforced connection” between ALS and SMA highlighted in the Abstract and the in Discussion should be toned down.
2. If FUS’s ZnF domain binds to the Sm site and effectively block the assembly of the Sm protein ring, leading to defective spliceosome function as the authors report, would that not be a toxic loss of function, rather than a toxic gain of function mentioned at the bottom of page 3 in the context of recent mouse experiments?
3. In Figure 1c, could the authors please explain the difference between “WT” and “FUS” in this western blot. Also, please show the complete gel, currently there appears to be three pieces which are out of order.
4. Bottom of page 6, In addition to the references 11 and 24 cited by the authors, reference 48 also previously reported that FUS binds U1 snRNP and both are mislocalized in the cytoplasm. So the authors’ finding here using the CLIP approach that FUS binds and aberrantly mislocalizes the U1 snRNP to the cytoplasm is not novel. What is novel and interesting, in my opinion, is their very detailed atomic level study of the binding interaction itself and the interesting competition with the Sm proteins.
5. Because of the relatively large size of the U1 snRNA, would one have expected to find a gel shift larger

than that observed in Figures 1E and 1F? Might there be a way to include size markers to estimate the MW of the bound RNAs?

6. Page 17. A-It could be helpful to include the sequence of the Twin Strep tag (WSHPQFEK-GGGSGGGSGGS-WSHPQFEK). 6B. Since the tag contains 4 aromatic residues and 4 cationic residues which tend to be common in RNA binding proteins, the authors ought to consider reporting a control experiment to test to see if the tag binds RNA. 6C. In addition, since the tag is placed at the C-terminus, could it interfere with the NLS ? This could be an important point considering that mutations in the NLS are known to cause FUS is aberrantly accumulate in the cytoplasm.

7. Page 18, In this paper, the authors have highlighted the union of FUS with U1 snRNA, but it seems from the data in Figure 2 panel F that six or seven other RNAs bind within an order of magnitude as well as U1 snRNA. I appreciate that the authors have already done quite alot of work in this very detailed study, so additional experiments would be outside the scope of this MS, but nevertheless could they comment on the biological and pathological significance of these other binders? Do they also contain stem-loops? Regarding this panel's y-axis, it is a little weird that the ticks are plotted as factors of ten but the y-axis label reads "log 2 fold enrichment".

8. The NMR study is really first rate-- Bravo! Figure 3, panel H is quite attractive.

9.A. Page 20, panel C (HMQC spectra). Here it would be useful to show a few more spectra recorded at different FUS-RBD : U1 snRNP ratios to allow the reader to ascertain if binding takes place o slow or fast time scale.

9.B. There is one spectroscopic difference in the binding of U1 snRNP versus SL3 to FUS, namely the signal at 0.82 ppm (¹H) x 12.5 ppm (¹³C). Is this a Ile delta methyl? Is this difference worth commenting on?

10. In Figure 4C (right), could the authors explain why there is no heat change observed for the RRM domain binding to SL3? It is understood that the mutation blocks the binding of the ZnF to the Sm site, but there still should be a heat change from the RRM/SL3 interaction.

11. Figure 4, panel D, a control chromatogram of the snRNP alone (without FUS or the Sm proteins) should be included here.

12. Page 25. The authors mention that the E. coli cultures were moved towards M9 minimal media with D2O. What was the final concentration of D2O for expression ?

13. Page 26. Change "was" to "were" on line 3, as two different types of spectra were recorded.

14. Page 27. In the paragraph "Size exclusion.." 1st line.. Is the word "increase" really necessary?

15. Page 28. The representation of the calorimetry results is a little odd. Perhaps this could be better: $(1.25 \pm 0.08) \cdot 10^5$ (molar?) (Are the units molar?) and $(2.09 \pm 0.08) \cdot 10^4$ kcal/mol
16. Page 29. Consider removing the parenthesis around (5% O₂) and mind the subscript on CO₂
17. Page 30. (middle) ..boiling.. at 70°C.. (how high is Zurich? ;-) maybe “..heating.. at 70°C...” would be better.
18. Page 33. What do the authors mean by “head over tail incubation”?
19. Page 35. The treatment used, namely, xylene, ethanol and boiling in the microwave at high power for 20 minutes, seems to be harsh. Could this damage the antigens that the authors are trying to retrieve ?
20. Pages 35 - 40, there are a series of words (e.g. Xylene, Formamide, Imidazol, Glycerol, Methanol, Acetic acid, PolyL-ornithine, Retinoic acid, Laminin) which are capitalized but should not be.
21. Page 40. There is a misplaced period on the second line of the second paragraph.
22. Page 41. Second line. The “2” in “cm²” should be superscripted.
23. The methods section is long and very detailed, which is good, however for readability and size considerations, some details could be moved to the Supp. Materials.
24. In this study, the authors have examined several different classes of non-coding RNAs for possible interactions with FUS, but they do not mention “tiRNAs”. Therefore, do their results rule out possible interactions with tiRNA, which are tRNA fragments formed by the action of the selective RNase Angiogenin. These tiRNAs have been reported to promote a healthy stress granule response by Ivanov P et al. Mol. Cell. 43: 613-23 & PNAS-USA 111: 18201-6.

Reviewer #2 (Remarks to the Author):

In this study, the Ruepp and Allain groups teamed up to identify and characterize an unusual interaction of the FUS protein with U1snRNA. Using a combination of advanced CLIP technologies and state-of-the-art NMR, they show that the RRM of FUS contacts U1 via stem/loop 3. Furthermore, cytosolic ALS-linked FUS has the same RNA target but is shown to interact with U1 in an aberrant way through its zinc finger region. Importantly, this disease-related interaction interferes with the assembly of U1 snRNP with the Sm proteins and traps the assembly pathway in vivo. The results of this manuscript therefore provide evidence for a toxic gain of function of FUS in ALS and raise the interesting hypothesis, that the etiologies of ALS and SMA, a neuromuscular disease linked to defects in snRNP assembly share common

aspects.

This is a well-written manuscript with convincing and clean data that sheds interesting and important light on the cellular function of FUS and also on its role in ALS. The study is hence of relevance for biochemists and biomedical researches alike. I have only a few remarks/criticisms that should be addressed by the authors:

The CLIP shown in Fig.2 not only identifies U1 but also other snRNAs (mostly U2 and U4) as primary targets for FUS. Does this reflect a similar mode of interaction with FUS as shown for U1 and how does the ALS-FUS version behaves? I guess the authors have already these data in their CLIP datasets and hence they should comment on it.

Does the FUS-ALS induced trapping of the assembly pathway also leads to an accumulation of snRNP assembly factors (SMN, Gemins, pICln)? It might well be that this has been shown already elsewhere but the authors should comment on this and/or provide the relevant data.

Minor point:

SMN stands for "survival motor neurons" (page 13)

Reviewer #3 (Remarks to the Author):

Autosomal dominant mutations in FUS result in neuronal and glial FUS inclusions and early-onset ALS. In this report, Jutzi and colleagues investigate FUS RNA targets initially using SH-SY5Y CLIPseq with region, and ALS mutation, -specific FUS constructs (FL FUS, Δ LC-FUS, Δ LC-FUSP525L). CIMS analysis, which resulted in the identification of U1 SL3 as the top FUS snRNA binding site, was confirmed by NMR of FUS-RBD (RRM, RGG2, ZnF) interactions with U1 snRNP and the relevant chemical shifts were also obtained using only SL3 RNA. Using the Δ LC-FUSP525L mutant, they show that cytoplasmic FUS targets both U1 SL3 and the U1 Sm site. NMR and core snRNP assembly in vitro experiments provided evidence that the RRM and ZnF interact with U1 SL34 containing SL3 and the Sm site and FUS ZnF-U1 Sm site interactions block Sm ring assembly. Stress induction of isogenic WT, KO and P5252L iPSC-derived MNS showed co-localization of FUS condensates with U1 snRNP and RNA FISH of an ALS splice site mutant mouse model, FUS Δ 14, revealed increases in cytoplasmic and nucleolar U1 snRNA suggesting abnormal snRNP biogenesis. Overall, the result is an insightful, comprehensive and thought-provoking study that identifies FUS specific interactions with U1 snRNA SL3 and indicates that snRNP biogenesis may be impaired in ALS suggesting mechanistic similarities with SMA. I have only a few comments that should be addressed.

1. Fig. 1c. Compared to WT, both Δ LC-FUS and Δ LC-FUSP525L appear to be expressed at a higher level so quantification of these expression levels should be included.

2. Fig. 2b. The Δ LC-FUS read scales are $\sim 10X$ and $\sim 3X$ higher for MALAT1 and HNRNPA2B1, respectively, compared to FUS. What is the explanation for the enhanced crosslinking of this C-terminal region and its re-distribution to HNRNPA1 coding exons? Is the LC region required to inhibit pervasive interactions of FUS with RNAs?

3. Fig. 5.

a. Fig. 5b,c. SA treatment is used to induce FUS condensates but do the P525L/P525L condensates co-localize with SGs? If so, the specificity of FUS-U1 snRNA co-localization in Fig. 5c is questionable. On a more minor point, I understand the authors are using SA to trigger FUS cytoplasmic condensates that also occur in ALS, but this treatment should not be interpreted as 'a mimic of the adult disease'.

b. Fig. 5c. What is the explanation for the U1 snRNA nuclear re-distribution between the WT and the FUS KO? For the P525L/P525L mutant, FUS distribution also appears to be more focal with less nuclear signal compared to Fig. 5B

4. Discussion.

a. The authors should mention a recent study (PMID: 32691043) that supports roles for K315/K316 (RRM) and K510 (NLS) lysine acetylation in the regulation of FUS RNA binding and localization, respectively.

b. An earlier report using humanized FUS mice (PMID: 32325059) showed that ALS-linked mutations activate an ISR, suppress axonal translation and result in progressive motor disease in the absence of aberrant splicing of FUS-associated pre-mRNAs. How do the authors interpret this observation in light of their U1 snRNP findings?

Point-by-point response to the comments of the reviewers

We are very grateful to the editor and the reviewers for their insightful comments and suggestions that have helped us to improve the quality of the manuscript. Below, we provide a point-by-point response in orange to their comments in black. Changes in the manuscript in response to comments and editorial guidelines are highlighted in red.

Reviewer #1 (Remarks to the Author):

The manuscript “Aberrant interactions of FUS...” NCOMMS-20-23834, is a real tour-de-force combining several different cutting edge techniques. In my opinion, it certainly meets the Nat. Comm. journal requirements of being technically sound, presenting strong evidence for the conclusions and presenting novel findings which stimulate new directions in the field. In particular, the solution structure of the FUS RBD + U1 snRNP SL3 RNA complex and cell fluorescence microscopy studies make a strong case that splicing factors like FUS and TDP-43 do not just bind mRNAs but can bind to other nuclear RNAs and aberrantly transport them outside nucleus.

Minor comments:

1. Abstract and Discussion (page 13). In my opinion, the connection between spinal muscular atrophy / survival of motor neuron protein and ALS / FUS + U1 snRNA is not so clear as the authors do not seem to have reported any results on the survival of motor neuron protein. Therefore, it seems that the “reinforced connection” between ALS and SMA highlighted in the Abstract and the in Discussion should be toned down.

We agree with the reviewer that we did not report any direct results on SMN. However, we don't think that the connection is reinforced, given that several studies have already indicated links between the two motor neuron diseases. To give the reader better insights into these previously described connections, we introduced a new paragraph into the discussion that briefly describes these relevant findings and puts our study more into context.

2. If FUS's ZnF domain binds to the Sm site and effectively block the assembly of the Sm protein ring, leading to defective spliceosome function as the authors report, would that not be a toxic loss of function, rather than a toxic gain of function mentioned at the bottom of page 3 in the context of recent mouse experiments?

We believe there is a misunderstanding here. In this context, the term “gain-of-function” refers to a toxic activity of the FUS protein in the cytoplasm. We propose that the aberrant interaction between mutant FUS and the U1 snRNA represents such a toxic gain-of-function, which in turn could lead to a loss-of-function of the spliceosome.

3. In Figure 1c, could the authors please explain the difference between “WT” and “FUS” in this western blot. Also, please show the complete gel, currently there appears to be three pieces which are out of order.

WT refers to the parental wild-type SH-SY5Y cell line that we used to delete the endogenous FUS gene and introduce our TS-tagged FUS CLIP constructs. We adapted the labelling in figure 1c and the figure legend to clarify this point.

The western blot in Figure 1c is cropped to prevent confusion because the normaliser Tub1A2 was visualised in the same channel as FUS and runs between the full-length and truncated FUS constructs. However, for transparency, we generated a new supplemental figure, where we present all our uncropped western blots.

4. Bottom of page 6, In addition to the references 11 and 24 cited by the authors, reference 48 also previously reported that FUS binds U1 snRNP and both are mislocalized in the cytoplasm. So the authors' finding here using the CLIP approach that FUS binds and aberrantly mislocalizes the U1

snRNP to the cytoplasm is not novel. What is novel and interesting, in my opinion, is their very detailed atomic level study of the binding interaction itself and the interesting competition with the Sm proteins.

We agree that the novelty in our manuscript comes from the molecular and structural insights into the details of the FUS – U1 snRNP interaction and we believe that our text is carefully written to properly acknowledge previous contributions. Of course, we are happy to include additional references that may have slipped our attention. Instead of reference 48, we think that reference 25 (by the same authors) is more appropriate here.

5. Because of the relatively large size of the U1 snRNA, would one have expected to find a gel shift larger than that observed in Figures 1E and 1F? Might there be a way to include size markers to estimate the MW of the bound RNAs?

In our CLIP experiments, the crosslinked RNAs were partially digested to yield fragments of about 100 nucleotides in size (therefore CLIP reads provide positional information about the FUS-binding site). Hence, the molecular weight shift observed in the autoradiographs is independent of the original transcript size.

6. Page 17. A-It could be helpful to include the sequence of the Twin Strep tag (WSHPQFEK-GGGSGGGSGGS-WSHPQFEK). 6B. Since the tag contains 4 aromatic residues and 4 cationic residues which tend to be common in RNA binding proteins, the authors ought to consider reporting a control experiment to test to see if the tag binds RNA. 6C. In addition, since the tag is placed at the C-terminus, could it interfere with the NLS? This could be an important point considering that mutations in the NLS are known to cause FUS to aberrantly accumulate in the cytoplasm.

A) We believe that the sequence of the Twin Strep tag is a detail for the interested reader and therefore included it in the plasmid section of the supplemental methods.

B) In Supplementary Figure 5e, we show electrophoretic mobility shift assays with Twin Strep-tagged WT and RNA-binding mutant FUS-RBD constructs. As the mutations in the RNA-binding domains almost fully abrogate RNA-binding, we conclude that the Twin Strep tag does not significantly interact with RNA. Furthermore, the residual binding activity observed by the RNA-binding mutant at high concentrations is more likely to be mediated by the RGG2 domain of FUS which is net positively charged and known to interact with RNA (Loughlin et al., 2019).

C) In Figures 1d-e and Supplementary Figure 1c, we show immunofluorescence images of the C-terminally Twin Strep tagged FUS constructs confirming that they do localise to the nucleus. Hence, the tag does not interfere with the function of the NLS.

7. Page 18, In this paper, the authors have highlighted the union of FUS with U1 snRNA, but it seems from the data in Figure 2 panel F that six or seven other RNAs bind within an order of magnitude as well as U1 snRNA. I appreciate that the authors have already done quite a lot of work in this very detailed study, so additional experiments would be outside the scope of this MS, but nevertheless could they comment on the biological and pathological significance of these other binders? Do they also contain stem-loops? Regarding this panel's y-axis, it is a little weird that the ticks are plotted as factors of ten but the y-axis label reads "log 2 fold enrichment".

The finding that additional snRNAs were also highly enriched in the FUS CLIP data is intriguing. Indeed, we found that the crosslinks between FUS and these other snRNAs occur at defined regions that are solvent exposed in the spliceosomal pre-B and B complexes. Given that these interactions either occurred at a stem-loop (U4), internal loop (U5) or a YNY-containing motif (U6), we hypothesized that all snRNAs are directly contacted by the RRM domain of FUS. By performing NMR spectroscopy with our FUS-RRM construct and *in vitro* transcribed U4, U5 and U6 snRNA fragments, we could confirm this hypothesis. These exciting new findings are described in the main text and presented as supplementary figure 5.

Regarding figures 2e and f, the plots were generated using the R package "ggplot2" which labels the y-axis as factors of 10 by default. However, to prevent confusion we changed the labels of the y-axis

to factors of 2.

8. The NMR study is really first rate-- Bravo! Figure 3, panel H is quite attractive.

Many thanks for this comment, we appreciate it.

9.A. Page 20, panel C (HMQC spectra). Here it would be useful to show a few more spectra recorded at different FUS-RBD : U1 snRNP ratios to allow the reader to ascertain if binding takes place on a slow or fast time scale.

Unfortunately, to keep the figure readable, we added a single intermediate point for each NMR titration shown in Page 20, panel C. However, these additional spectra recorded at ratio 1:0.5 (FUS-RBD:U1snRNP or FUS-RBD:SL3) are sufficient to show that the binding takes place in fast/intermediate exchange.

9.B. There is one spectroscopic difference in the binding of U1 snRNP versus SL3 to FUS, namely the signal at 0.82 ppm (¹H) x 12.5 ppm (¹³C). Is this a Ile delta methyl? Is this difference worth commenting on?

The spectroscopic difference in the binding of U1 snRNP versus SL3 to FUS pointed by the reviewer is due to the way the data are displayed (just a matter of levels). In the new panel C page 20, the levels of the spectra have been adjusted and this difference is not observed anymore.

10. In Figure 4C (right), could the authors explain why there is no heat change observed for the RRM domain binding to SL3? It is understood that the mutation blocks the binding of the ZnF to the Sm site, but there still should be a heat change from the RRM/SL3 interaction.

The bipartite binding of FUS (RRM+ZnF) on the SL34 RNA has a dissociation constant of 70 nM and the concentrations of RNA and protein have been optimized to observe the nanomolar binding using ITC. In this case, a protein solution concentrated at 60 μM is injected into a 6 μM RNA solution to obtain the curve shown in Fig 4c (left panel). The mutation of the GGU motif from the Sm site impairs bipartite binding. In this scenario, we expect to observe the binding of the RRM on stem loop 3 only. The conditions used to perform the ITC titration of FUS-RBD/SL34mut were the same than for the wt in order to be able to compare. However, using these conditions, binding occurring with a KD in the range of 10 μM cannot be observed. In order to monitor this interaction using ITC, one would need to strongly increase the amount of protein (to 600 μM) and RNA (50 μM) as we did for the ITC titration of FUS-RRM with SL3 (Supplementary Figure 3c). If one looks carefully at the Fig 4c (right panel), one can see that there is still a weak heat change with the RNA mutant.

11. Figure 4, panel D, a control chromatogram of the snRNP alone (without FUS or the Sm proteins) should be included here.

We wonder whether there is a misunderstanding here. In our experiment, the "snRNP alone" (without Sm proteins) would correspond to the naked SL34 RNA, as the U1-specific proteins are not part of our minimalistic core snRNPs. We think that a control chromatogram of the SL34 RNA alone will complexify the figure without bringing any add-on.

However, we provide for the reviewer, a figure comparing the chromatograms of U1SL34, U1SL34+FUS-RBD and U1SL34+Sm Core:

Figure legend: A- Chromatogram showing the OD at 260 nm as a function of the elution volume when U1 SL34 was loaded on the size exclusion column S200 increase. B- Chromatogram showing the OD at 260 nm as a function of the elution volume when U1 SL34-Sm core was loaded on the size exclusion column S200 increase. C- Chromatogram showing the OD at 260 nm as a function of the elution volume when U1 SL34-FUSRBD was loaded on the size exclusion column S200 increase. D- Overlay of the three chromatograms shown in A, B and C.

This figure shows that the elution volumes of the complexes (Sm core-SL34 or FUSRBD-SL34) is smaller than the elution volume of the free SL34, in agreement with an increase of the hydrodynamic radii of the protein-RNA complexes when compared to the free RNA.

12. Page 25. The authors mention that the E. coli cultures were moved towards M9 minimal media with D2O. What was the final concentration of D2O for expression?

The M9 minimal medium used to express the ILV FUS-RBD sample was prepared in 99.8%D2O in presence of protonated glucose (2g/L).

13. Page 26. Change “was” to “were” on line 3, as two different types of spectra were recorded.

14. Page 27. In the paragraph “Size exclusion..” 1st line.. Is the word “increase” really necessary?

15. Page 28. The representation of the calorimetry results is a little odd. Perhaps this could be better: $(1.25 \pm 0.08) \cdot 10^5$ (molar?) (Are the units molar?) and $(2.09 \pm 0.08) \cdot 10^4$ kcal/mol

16. Page 29. Consider removing the parenthesis around (5% O₂) and mind the subscript on CO₂

13 - 16) Thank you for your good eye. We adapted the manuscript accordingly.

17. Page 30. (middle) ..boiling.. at 70°C.. (how high is Zurich? ;-) maybe “..heating.. at 70°C...” would be better.

We completely agree with the reviewer. This experiment was performed in Bern (540 m), which is at a higher altitude than Zurich (408 m), but indeed this is not at a high enough altitude to get the samples boiling at the temperature. :) We adjusted the text according to your suggestion.

18. Page 33. What do the authors mean by “head over tail incubation”?

The term refers to incubation on a rotating wheel. We clarified that in the text accordingly.

19. Page 35. The treatment used, namely, xylene, ethanol and boiling in the microwave at high power for 20 minutes, seems to be harsh. Could this damage the antigens that the authors are trying to retrieve?

Compared to the methods used for cell lines, the procedure to retrieve antigens in tissue indeed seems harsh. However, heat induced epitope retrieval as employed in this study is a standard protocol for paraffin-embedded formalin fixed tissue (E.g. see <https://www.abcam.com/protocols/ihc-antigen-retrieval-protocol>). It was also applied for IHC on FUS Δ 14 mouse sections before in Devoy et al, 2017, PMID: 29053787.

20. Pages 35 - 40, there are a series of words (e.g. Xylene, Formamide, Imidazol, Glycerol, Mathanol, Acetic acid, PolyL-ornithine, Retinoic acid, Laminin) which are capitalized but should not be.

21. Page 40. There is a misplaced period on the second line of the second paragraph.

22. Page 41. Second line. The “2” in “cm²” should be superscripted.

20 – 22) Again, thank you for spotting these mistakes, which we corrected.

23. The methods section is long and very detailed, which is good, however for readability and size considerations, some details could be moved to the Supp. Materials.

We agree with this comment and note that the word count of our methods section (6,555) is clearly above the typical word count of 3,000 that is recommended by the journal. We therefore moved the sections describing standard procedures as well as preparatory methods or side experiments to the supplemental material. This reduced the word count in the manuscript to 4,400.

24. In this study, the authors have examined several different classes of non-coding RNAs for possible interactions with FUS, but they do not mention “tiRNAs”. Therefore, do their results rule out possible interactions with tiRNA, which are tRNA fragments formed by the action of the selective RNase Angiogenin. These tiRNAs have been reported to promote a healthy stress granule response by Ivanov P et al. Mol. Cell. 43: 613-23 & PNAS·USA 111: 18201-6.

In this manuscript, we describe how FUS binds to the snRNAs because they were the most enriched biotype in our CLIP analysis and contain the top FUS interactor – the U1 snRNA. Given that FUS regulates pre-mRNA splicing in various cell types and organisms, it is very likely that these interactions with snRNAs are functionally important, which is why we focused on them. However, given the low RNA-binding specificity of FUS *in vivo*, we also observed FUS binding to other types of abundant non-coding RNAs like snoRNAs, Y RNAs and tRNAs. In the case of tRNAs, the reads indeed mostly map to the 5'-half of the genes (see selected examples a-c below), whereas some reads cover the whole length of tRNA genes (see example d). Unfortunately, CLIP cannot discriminate between FUS binding to tRNA fragments and full-length tRNAs due to the RNase1 digestion step. If RNase1 cuts tRNAs in their anti-codon loops, the resulting RNA fragment looks like a tRNA half, even though it has been generated post-lysis. In summary, albeit our data cannot provide information about FUS interactions with tiRNAs, it certainly does not rule them out.

Reviewer #2 (Remarks to the Author):

In this study, the Ruepp and Allain groups teamed up to identify and characterize an unusual interaction of the FUS protein with U1snRNA. Using a combination of advanced CLIP technologies and state-of-the-art NMR, they show that the RRM of FUS contacts U1 via stem/loop 3. Furthermore, cytosolic ALS-linked FUS has the same RNA target but is shown to interact with U1 in an aberrant way through its zinc finger region. Importantly, this disease-related interaction interferes with the assembly of U1 snRNP with the Sm proteins and traps the assembly pathway *in vivo*. The results of this manuscript therefore provide evidence for a toxic gain of function of FUS in ALS and raise the interesting hypothesis, that the etiologies of ALS and SMA, a neuromuscular disease linked to defects in snRNP assembly share common aspects.

This is a well-written manuscript with convincing and clean data that sheds interesting and important light on the cellular function of FUS and also on its role in ALS. The study is hence of relevance for biochemists and biomedical researches alike. I have only a few remarks/criticisms that should be addressed by the authors:

The CLIP shown in Fig.2 not only identifies U1 but also other snRNAs (mostly U2 and U4) as primary targets for FUS. Does this reflect a similar mode of interaction with FUS as shown for U1 and how does the ALS-FUS version behaves? I guess the authors have already these data in their CLIP datasets and hence they should comment on it.

Thank you for picking up on this interesting point. As described above, we found that the RRM of FUS specifically interacts with the U4, U5 and U6 snRNAs and included the relevant data in our manuscript. Even though the U2 snRNA is highly enriched in our data, the FUS cross-links were not clustered in a specific region of the transcript, arguing against a specific recognition. In the cytoplasmic FUS CLIP, we also detected binding to these additional snRNAs, but failed to identify cross-links to their Sm sites. This could be explained by the fact that the U1 Sm site harbours a GGU motif (which is recognised the ZnF domain of FUS), whereas the other Sm sites lack this motif. Therefore, the competition between FUS and Sm proteins is likely to be specific for the U1 snRNA.

Does the FUS-ALS induced trapping of the assembly pathway also leads to an accumulation of snRNP assembly factors (SMN, Gemins, piCln)? It might well be that this has been shown already elsewhere but the authors should comment on this and/or provide the relevant data.

There are indeed publications describing aberrant interactions between FUS and SMN in the cytoplasm (Yamazaki et al., 2012 and Sun et al., 2015) and showing that cytoplasmic FUS condensates co-localise with SMN (Groen et al., 2013). We have now mentioned that in the discussion.

Minor point:
SMN stands for “survival motor neurons” (page 13)

Thank you for pointing that out. We have adapted that accordingly.

Reviewer #3 (Remarks to the Author):

Autosomal dominant mutations in FUS result in neuronal and glial FUS inclusions and early-onset ALS. In this report, Jutzi and colleagues investigate FUS RNA targets initially using SH-SY5Y CLIPseq with region, and ALS mutation, -specific FUS constructs (FL FUS, Δ LC-FUS, Δ LC-FUSP525L). CIMS analysis, which resulted in the identification of U1 SL3 as the top FUS snRNA binding site, was confirmed by NMR of FUS-RBD (RRM, RGG2, ZnF) interactions with U1 snRNP and the relevant chemical shifts were also obtained using only SL3 RNA. Using the Δ LC-FUSP525L mutant, they show that cytoplasmic FUS targets both U1 SL3 and the U1 Sm site. NMR and core snRNP assembly in vitro experiments provided evidence that the RRM and ZnF interact with U1 SL34 containing SL3 and the Sm site and FUS ZnF-U1 Sm site interactions block Sm ring assembly. Stress induction of isogenic WT, KO and P525L iPSC-derived MNs showed co-localization of FUS condensates with U1 snRNP and RNA FISH of an ALS splice site mutant mouse model, FUSDelta14, revealed increases in cytoplasmic and nucleolar U1 snRNA suggesting abnormal snRNP biogenesis. Overall, the result is an insightful, comprehensive and thought-provoking study that identifies FUS specific interactions with U1 snRNA SL3 and indicates that snRNP biogenesis may be impaired in ALS suggesting mechanistic similarities with SMA. I have only a few comments that should be addressed.

1. Fig. 1c. Compared to WT, both Δ LC-FUS and Δ LC-FUS-P525L appear to be expressed at a higher level so quantification of these expression levels should be included.

The truncated FUS constructs are indeed expressed at a higher level, but the overexpression is below 2-fold compared to the endogenous FUS protein. We have now included a relative quantification in figure 1c.

2. Fig. 2b. The Δ LC-FUS read scales are ~10X and ~3X higher for MALAT1 and HNRNPA2B1, respectively, compared to FUS. What is the explanation for the enhanced crosslinking of this C-terminal region and its re-distribution to HNRNPA1 coding exons? Is the LC region required to inhibit pervasive interactions of FUS with RNAs?

There is a technical explanation for these differences in read numbers: The Δ LC-FUS CLIP was analysed in a separate sequencing experiment that was much deeper (130 mio reads per replicate) than the other two CLIP experiments (30 mio reads per replicate). Therefore, the absolute read numbers cannot be directly compared in this case. However, we see a clear loss of intronic binding in the HNRNPA2B1 pre-mRNA. We believe that this could be explained by the fact that phase separation is required for FUS to associate with chromatin, where pre-mRNA splicing takes place. Mechanistically, this could depend on direct interactions with hnRNPs or other chromatin associated factors. To clarify that, we included a sentence and corresponding reference on page 6.

3. Fig. 5.

a. Fig. 5b,c. SA treatment is used to induce FUS condensates but do the P525L/P525L condensates co-localize with SGs? If so, the specificity of FUS-U1 snRNA co-localization in Fig. 5c is questionable. On a more minor point, I understand the authors are using SA to trigger FUS cytoplasmic condensates that also occur in ALS, but this treatment should not be interpreted as ‘a mimic of the adult disease’.

A) In supplementary figure 8b, we show that the FUS-P525L condensates co-localise with the stress granule marker TIAR. Given that upon stress the U1 snRNA (and the other snRNAs) also mis-localise in FUS KO cells, we conclude that the trapping of snRNAs is a non-specific effect. We therefore advocate a model, where snRNAs are non-specifically trapped in stress granules, which are then bound by ALS-associated FUS and could mature into cytoplasmic inclusions. We agree that the statement “to mimic the adult disease” is too strong and therefore removed it.

b. Fig. 5c. What is the explanation for the U1 snRNA nuclear re-distribution between the WT and the FUS KO? For the P525L/P525L mutant, FUS distribution also appears to be more focal with less nuclear signal compared to Fig. 5B

B) Our explanation is that the U1 snRNA is trapped in stress granules in a FUS-independent manner. This agrees with our observations for the other snRNAs and Snurportin-1 (supplementary figure 8) and is consistent with our model (figure 6b).

In figure 5c, the FUS P525L distribution appears more focal and less nuclear because the images were acquired at higher resolution on a confocal microscope, whereas the images in figure 5b are non-confocal.

4. Discussion.

a. The authors should mention a recent study (PMID: 32691043) that supports roles for K315/K316 (RRM) and K510 (NLS) lysine acetylation in the regulation of FUS RNA binding and localization, respectively.

b. An earlier report using humanized FUS mice (PMID: 32325059) showed that ALS-linked mutations activate an ISR, suppress axonal translation and result in progressive motor disease in the absence of aberrant splicing of FUS-associated pre-mRNAs. How do the authors interpret this observation in light of their U1 snRNP findings?

A) We agree that this study, which was published just a few days after our initial submission is relevant to our manuscript. Therefore, we included a sentence to mention these findings in the discussion of our FUS – stem-loop 3 structure.

B) In combination with other publications, the report by Lopez-Erauskin et al. strengthens the idea that a loss of nuclear FUS function is not a main contributor to the disease. Consequently, the mice show a disease phenotype in the absence of aberrant splicing of FUS-dependent pre-mRNAs. However, this result is compatible with our gain-of-function model, where ALS-associated FUS induces alterations in snRNP homeostasis that ultimately affect the splicing or polyadenylation of pre-mRNAs that are **not** regulated by FUS in the nucleus. Indeed, another recent mouse study identified splicing alterations that only occur in the presence of cytoplasmic FUS. Intriguingly, one of the affected pre-mRNAs (SmB/B') is known to be sensitive to levels of the core splicing machinery. We think that these published results reinforce our hypothesis and therefore we included a new sentence in the discussion.

REVIEWERS' COMMENTS

Reviewer #1 (Remarks to the Author):

The authors have done a very thorough job revising the MS and have satisfied all my concerns.

Reviewer #2 (Remarks to the Author):

The authors have addressed the criticisms/suggestions raised by all three reviewer in a satisfactory manner. I therefore support publication of this interesting work in NatComm.

Reviewer #3 (Remarks to the Author):

The authors have provided a thoughtful and thorough response to all of the prior comments and I have no remaining issues.